

# Forcing-dependent submesoscale variability and subduction in a coastal sea area (Gulf of Finland, Baltic Sea)

Kai Salm, Germo Väli, Taavi Liblik, and Urmas Lips

Department of Marine Systems, Tallinn University of Technology, Tallinn, Estonia.

*Correspondence to*: Kai Salm (Kai.Salm@taltech.ee)

**Abstract.** Submesoscale (SMS) processes within a stratified coastal environment are characterized based on glider missions and a realistic simulation with the SMS permitting grid spacing of 0.125 nautical miles. The study period covered the conditions of the developing and established seasonal stratification in the Gulf of Finland (Baltic Sea). The tracer variance maximum was detected around the upper mixed layer (UML) depth and shallower than the depth of the maximum density

gradient in spring and in the seasonal thermocline in late summer. We suggest that atmospheric forcing is the key driver of these disparities. In spring and early summer, predominantly positive net surface heat flux promoted water column stratification, and wind-induced mixing often did not reach the lower portion of the UML – likely SMS flows due to the horizontal buoyancy gradients developed there. In conditions, when the net surface heat flux was close to zero or negative, wind forcing had a more prominent role in shaping dynamics within the thermocline. High SMS tracer variance was

consistently observed and simulated in the subsurface layers at the offshore side of a coastal baroclinic current characterized by slanted isopycnals. We propose that topography-related instabilities of frontal currents can create favourable conditions for SMS subduction transporting tracers from the sea surface near coastal boundary towards offshore and downwards, below the seasonal thermocline.

## 1 Introduction

The ocean is a vast dynamic system that is constantly in motion, driven by forces, including winds, tides, and changes in temperature and salinity. As a result, a range of different scales of motion forms. In a common theoretical framework, upper ocean's turbulence is dominated by the mesoscale quasi-geostrophic eddies, internal waves, and microscale three-dimensional turbulence, but recently the attention has been on the contribution of the submesoscale processes (e.g., Taylor and Thompson, 2023). The submesoscale (SMS) is recognized to transfer energy from the geostrophically near-balanced

flows to small-scale turbulence (Capet et al., 2008; Naveira Garabato et al., 2022).

At the mesoscale, i.e., characteristic horizontal scale of O(10–100) km in the ocean and O(5–20) km in the Baltic Sea, the flow is close to the geostrophic balance. Dynamically, mesoscale eddies are characterized by small Rossby number $\left(Ro = \frac{U}{fL}, \; Ro \ll 1\right)$ and large Richardson number $\left(Ri = \frac{N^2 H^2}{U^2}, \; Ri \gg 1\right)$, meaning rotational and buoyancy effects are strong in



shaping fluid motion. U is a characteristic horizontal velocity scale, L and H horizontal and vertical length scales, f the

Coriolis parameter, and N the Brunt-Väisälä frequency. The SMS occupies intermediate space scale of O(1) km and is

characterized with Ro and Ri both in the order of O(1) (Thomas et al., 2008), meaning rotation, stratification, and inertia are

all important to the SMS dynamics. Compared to the mesoscale, the Earth rotation does not constrain the motion to the same

extent and thus, strong vertical motions can develop. The SMS processes effectively induce the vertical fluxes of heat,

carbon and nutrients between the surface and the interior, making them relevant to phytoplankton productivity (Mahadevan,

35   2016).

The Baltic Sea is a shallow marginal sea in Northern Europe that stretches from 54° N to 66° N. The sea has limited water

exchange with the North Sea, and the input of fresh river water is large. The largest river discharge is located at the eastern

end of the Gulf of Finland (GoF), a sub-basin in the northeastern Baltic Sea. The gulf is characterized by the pronounced

horizontal salinity gradient, resulting from significant freshwater runoff at the eastern end of the gulf and the transport of

saltier water into the gulf through its western border. The flow field in the GoF, affecting the freshwater transport and

shaping horizontal and vertical salinity gradients, is significantly influenced by the wind (Lilover et al., 2017; Westerlund et

al., 2019). For instance, in the case of prevailing along-gulf winds, a circulation pattern with currents along the wind near the

coasts and in the opposite direction in the central gulf can develop (Lips et al., 2017; Elken et al., 2011) that, as a result,

enhances transverse salinity gradients.

In contrast to the open ocean, the vertical salinity gradient has an equally important role alongside temperature in shaping the

density stratification of the water column in the GoF. In spring–summer, a seasonal thermocline forms around depths of 10–

meters, while below it and in the whole water column during the rest of the year, the salinity governs the vertical

stratification. The quasi-permanent halocline lies below 60 meters on average (Liblik and Lips, 2017). During winter and

early spring, shallow haline stratification can develop because of freshwater advection (Liblik et al., 2020; Lips et al., 2017).

The along-gulf winds favour the occurrence of upwelling and downwelling events off the northern and southern coasts

(Kikas and Lips, 2016; Lehmann et al., 2012). The summer upwelling events are typically associated with substantial

temperature gradients at the sea surface (Lips et al., 2009; Uiboupin and Laanemets, 2009). Due to the pronounced

horizontal and vertical salinity gradients in the GoF and the development of jet currents along the upwelling fronts (Suursaar

and Aps, 2007), upwelling events also alter the salinity distribution. By analysing the variability and patterns of temperature

and salinity distributions, one can estimate the mixing (Lips et al., 2009) and SMS activity (Väli et al., 2017) associated with

upwelling events.

The increasing number of studies analysing the captured SMS variability in the Baltic Sea supports the prevalence of the

SMS processes (Carpenter et al., 2020; Lips et al., 2016a; Salm et al., 2023).  Although the high-resolution measurements

gathered with autonomous devices offer insights into the water column structure, the analysis regarding the SMS from the

observational data are limited. Therefore, complementing observations with numerical modelling yields supplementary data

that enhance our analytical capabilities and enables extrapolation of the findings in space and time. The models allow us to



describe the background forcing and estimate the quantities indicative of the SMS regime that cannot be derived from the glider mission data alone.

In this study, we aim to explore the role of SMS processes within a stratified coastal environment, focusing on the upper half of the water column. Three glider missions were conducted in the GoF in spring–summer 2018–2019. The appearance of SMS features favoured by the development of a mesoscale front and changing wind forcing was documented based on mission data from spring 2018 (Salm et al., 2023). We employ the data of all three missions from the same area by analysing tracer variability at the horizontal scale of a km. We compare the measurements with the simulation that has the SMS permitting grid spacing of 0.125 nautical miles (approximately 232 m) and use the model data to analyse the background forcing and dynamical properties, stratification, and coastal upwelling and downwelling events. Furthermore, model data allow us to extrapolate the analysis over the whole spring-summer season of both years.

We use spice – the temperature and salinity differences that cancel each other in the density – as a passive tracer and propose that the scale-dependent spice variance can be used as an indicator for the SMS dynamics (e.g., Rudnick and Cole, 2011). Spice can potentially be transported away from its source. However, with the high variability of the predominantly wind-driven current field in the GoF (Lilover et al., 2017) and the short-term nature of individual SMS features, this transport should be limited in space. Nevertheless, we relate the observed and modelled SMS variability to the conditions in a broader area to explain the emergence of high SMS spice variance at the background of mesoscale dynamics. Note also that using spice variance as an indicator of SMS activity is applicable best if the variability in temperature and salinity contribute almost equally to the variability in density, which is not the case at low temperatures (from late autumn to early spring) in the brackish waters of the GoF.

The following hypotheses are considered. We suggest that the variations in the intensity of submesoscale features depend on varying forcing and background (larger-scale) hydrographic conditions. Furthermore, we propose that topography-related instabilities of baroclinic coastal currents contribute to the vertical fluxes of tracers through the seasonal thermocline.

The paper is organized as follows: Section 2 introduces the observations, the numerical experiment, and the analysis methods; Section 3 provides the results, including the analysis of the spice variability and the SMS permitting events; Section 4 discusses the presence of the SMS activity in the study area; the last section summarizes the work.

## 2 Data and Methods

### 2.1 Glider observations

The study area is in the GoF, Baltic Sea (Fig. 1). Missions were carried out in May–June 2018 and 2019, and July–August 2019, during which an autonomous underwater vehicle, the Slocum G2 Glider MIA, collected oceanographic data along predefined transects. In May–June 2018, an 18 km long transect was sampled 26 times, and in 2019, 4.5 and 5.5 km long transects were sampled over 90 times. Altogether, over 12,000 profiles were gathered. The raw data were quality controlled, and the coefficients to account for the response time of the sensors and the thermal lag were defined to minimize differences




between two consecutive CTD profiles, similarly as described by Salm et al. (2023). The half YOs were bin-averaged to a
uniform 0.5 dbar vertical grid and arranged as profiles. For the analysis, the data fields were interpolated on a regular grid
with a time step of 10 min, which was equal to the horizontal distance of 130 m on average.

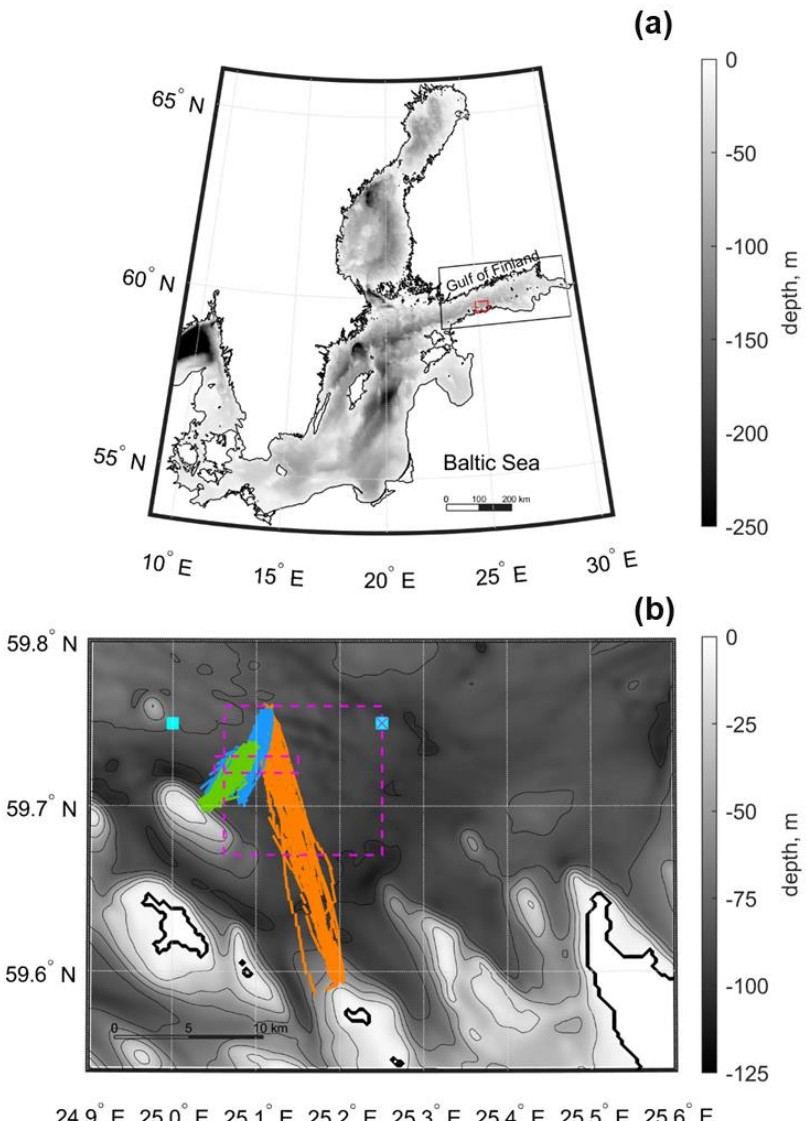

**Figure 1: Map of (a) the Baltic Sea and (b) the study area. The red rectangle on panel a shows the location of the study area. The depth isolines on panel b are presented with a step of 20 m. The glider missions carried out in May–June 2018 and 2019 and July–**
**August 2019 are shown in orange, blue, and green, respectively. The larger magenta box shows the "time series box" chosen for the modelled data, and the smaller one shows the common area covered in all missions (59.72–59.73° N 25.05–25.15° E) and used for the comparison of glider and model data. The cyan squares show the positions of the ERA5 grid points used for extracting the radiation data and the magenta cross the wind data. The colormap shows bathymetry used in the model.**



## 2.2 Model setup

We are using a three-dimensional nested setup of the General Estuarine Transport Model (GETM; Burchard and Bolding, 2002) to simulate the circulation and the temperature and salinity distribution in the GoF. GETM is a primitive-equation, free-surface, hydrostatic model with built-in vertically adaptive coordinates (Gräwe et al., 2015; Hofmeister et al., 2010; Klingbeil et al., 2018). Previous studies have shown that the total variation diminishing (TVD) advection scheme combined with the superbee limiter reduces the numerical mixing in the simulations (Gräwe et al., 2015; Klingbeil et al., 2018).

Vertical mixing in the GETM is calculated by coupling it with the General Ocean Turbulence Model (GOTM; Umlauf and Burchard, 2005) and more precisely, we are using the two-equation k−ε turbulence model (Burchard and Bolding, 2001; Canuto et al., 2001) in the simulation.

The entire GoF is the high-resolution model domain, with a uniform horizontal grid spacing of 0.125 nautical miles (approximately 232 m) and 60 adaptive layers in the vertical. Original bathymetry was obtained from the EMODnet database

(last accessed 09.06.2023) with a resolution of 1/16 arc minutes (approximately 150 m). Bathymetry data were averaged to the model resolution, and missing values were interpolated using the nearest neighbour (NN) technique.

The high-resolution simulation with an open western boundary at the GoF entrance was performed starting on 3 December 2017. The initial temperature and salinity fields and boundary conditions were taken from coarse-resolution simulations covering, respectively, the entire Baltic Sea with a grid step of 1 nautical mile (see Väli et al., 2024 for more details) and the

Baltic Proper (including the GoF) with a grid step of 0.5 nautical miles (see Zhurbas et al., 2018 and Liblik et al., 2020, 2022 for more details). As all setups use adaptive coordinates, we first interpolated profiles to the fixed 5 m vertical resolution before spatial interpolation to the high-resolution model grid using the NN method. If needed, the profiles were extended to the bottom of the high-resolution grid to compensate for the bathymetric differences. The model run started from a motionless state with zero sea surface height and current components. Previous studies by Krauss and Brügge (1991) and

Lips et al. (2016b) have shown that the spin-up time for the Baltic Sea model under atmospheric forcing is less than 10 days. For the boundary conditions, the temperature, salinity and current profiles and the sea surface height, all with 1-hour temporal resolution from the 0.5 nautical miles resolution simulation, were used. The same model setup was used by Siht et al. (in press), where further details and some validation results are presented.

The atmospheric forcing at the sea surface (the momentum and heat flux) is calculated from the European Centre for

Medium-Range Weather Forecasts atmospheric reanalysis data set (ERA5, Hersbach et al., 2020) by utilizing wind components and other relevant parameters (air temperature, total cloudiness, relative humidity, sea level pressure) for bulk formulae by Kondo (1975). All meteorological parameters are interpolated bi-linearly to the model grid. The riverine freshwater input to the Baltic Sea was taken from the dataset produced for the Baltic Model Intercomparison Project (BMIP; Gröger et al., 2022) based on the E-HYPE (Lindström et al., 2010) hindcast and forecast products by Väli et al. (2019).

There are 91 rivers in the dataset, of which 13 are in the GoF.



## 2.3 Submesoscale analysis

While air-sea fluxes, turbulent mixing, and advection introduce variability in the seawater properties, the dynamical processes act quickly to remove density differences. Spice – the temperature and salinity differences that cancel each other in density – is the trace that remains (Rudnick and Ferrari, 1999). Spice was defined as the sum of the temperature anomalies scaled by the thermal expansion coefficient and the salinity anomalies scaled by the haline contraction coefficient ($spice = \alpha \Delta T + \beta \Delta S$).

In this study, anomalies refer to the differences of measured or modelled Conservative Temperature or Absolute Salinity from the average value along an isopycnal calculated in the surrounding 4 km. For the observations, it was a moving window along the trajectory and for the model, it was a 4×4 km square. The length scale was chosen based on the internal Rossby deformation radius, which is typically 2–4 km in the GoF (Alenius et al., 2003).

We analysed the spice variance based on both the observations and the model results. Figure 2a shows an example of spice derived from the observations. The patches of positive (negative) spice indicate regions with higher salinity and warmer temperature (lower salinity and colder temperature) compared to the surrounding 4 km. Spice effectively highlights the smaller-scale (submesoscale) variability that may be challenging to discern solely from temperature and salinity distributions (Fig. 2b, c).

The average spice intensity was defined as the root mean square of spice from the sea surface to the depth of minimum temperature. The UML depth was determined as the minimum depth where $\rho_z \geq \rho_1 + 0.25 \, kg \, m^{-3}$ was satisfied ($\rho_z$ is the density at depth $z$ and $\rho_1$ at 1 m). The strength and position of the pycnocline (we use the term "upper pycnocline" (UP) for it) were analysed based on the maximum of $N^2$ and its depth.

For the model validation, we show that the development of stratification and tracer variability captured in the measurements is present in the simulation. The model data were averaged over a 1×1 km area, chosen inside a common area that was covered in all glider missions (59.72–59.73° N, 25.13–25.15° E, see Fig. 1). Considering submesoscale analysis, meaningful comparisons require averaging the model data over a 10×10 km study window, referred to as the "time series box" (59.67–59.76° N, 25.06–25.25° E; see Fig. 1), as features may be spatially and temporally displaced in the model.

The quantities indicative of the SMS regime, such as the Rossby number, balanced Richardson number, and horizontal buoyancy gradient, were calculated based on the model data. The Rossby number and the balanced Richardson number were estimated, respectively, as Eq. (1):

$$Ro = \frac{\zeta}{f}, \tag{1}$$

and Eq. (2):

$$Ri = \frac{f^2 N^2}{|\nabla_H b|^2}. \tag{2}$$





$\zeta = v_x - u_y$ is the vertical component of relative vorticity, $f$ Coriolis frequency, $N^2 = b_z$ vertical buoyancy gradient (squared Brunt-Väisälä frequency) and $|\nabla_H b| = \sqrt{b_x^2 + b_y^2}$ horizontal buoyancy gradient modulus. Buoyancy was defined as Eq. (3):

$$b = -\frac{g}{\rho_0}(\rho - \rho_0),\tag{3}$$

where $\rho_0$ is the reference density of 1000 kg m$^{-3}$. Central scheme was used to calculate the parameters from the model. Horizontal and vertical steps for estimating gradients were 500 m and 2 m, respectively.

Wind data and parameters defining heat exchange between the atmosphere and the sea surface were extracted on the ERA5 product grid cells covering the study area (59.50–59.75° N, 25.00–25.25° E; see Fig. 1). The wind components in the analysis were smoothed by a Gaussian low-pass filter for 6 h.

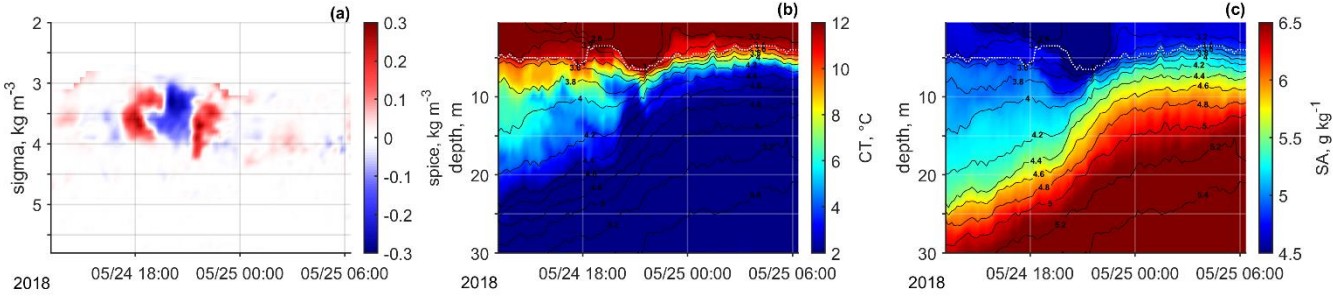


**Figure 2: An example of spice distribution (a) and temperature (b) and salinity (c) sections with overlapped density contours based on glider data from 24–25 May 2018. The white dashed line shows the upper mixed layer depth.**

### 3 Results

### 3.1 Stratification and spice variance from observations and model

First, we compared model results with the observed vertical structure of the water column and tracer variability. A dataset equivalent to the glider measurements was extracted from the model's 30-minute output. A common area covered in all missions was chosen (59.72–59.73° N 25.05–25.15° E, Fig. 1), and the consecutive profiles inside that area were averaged per transect. On average, this area covered a 1 km long glider transect, over which ten profiles were gathered.

The model captured the three-layered structure of the water column with the warm surface layer (upper mixed layer – UML),
cold intermediate layer (CIL), and saltier deep layer. The depth ranges between the UML and the CIL were 7–22 and 6–24 m (I mission), 13–39 and 12–29 m (II mission), 10–43 and 6–41 m (III mission) for the measurements and the model, respectively. However, the model tended to overestimate the temperature in the CIL and underestimate salinity, particularly in the deep layer (Fig. 3a–c). On average, the modelled salinity was 0.4 g kg$^{-1}$ lower than the measured salinity, and the CIL temperature was higher by about 2 and 1.5 °C than the measured temperature in 2018 and 2019, respectively.



Despite these discrepancies, the model replicated the characteristic layered structure of the water column in the GoF and the development of the seasonal thermocline (Fig. 4). Only some slight differences in the strength, structure and location of the seasonal thermocline can be indicated. The maximum vertical density gradient was at depths about 10 and 8 m (I mission), 20 and 15 m (II mission), and 14 and 9 m (III mission) based on the measurements and the model (Fig. 3). Thus, the model positioned the upper pycnocline at shallower depths than measured. Furthermore, the model did not capture the presence of

two observed local maxima of vertical density gradient in the upper part of the water column in May 2018 and July 2019 (Fig. 4).

The halocline was generally less pronounced in the model than in the observational data during all glider missions (Figs. 3–4). Since we focus on the seasonal thermocline (upper pycnocline – UP), we do not analyse this discrepancy here and assume that it does not impact the following results regarding the structure and variability in the UP.

Vertical distributions of spice were similar in the measurements and the simulation, although the maximum spice intensities were slightly lower in the model than in observations during both spring missions (Fig. 3d–f). The spice variability was the highest during the summer 2019 mission when the vertical temperature, salinity and density gradients were the largest and the vertical stratification was the strongest. During both spring missions, spice was the highest around the base of the UML and at shallower depths (lower densities) than the depth (density) of the maximum vertical density gradient (UP depth).

However, in July–August 2019, the spice peak was moved to the depths beneath the UML and was located immediately below the UP (strongest density gradient).







**Figure 3: The mission average temperature and salinity profiles (a–c). Red and blue correspond to temperature and salinity from the observations and dashed and solid black to temperature and salinity from the model simulations. The standard deviation of spice calculated from the measurements (black dots) and the modelled data (white circles) in each mission are shown on panels d–f. Columns I–III stand for May–June 2018, May–June 2019, and July–August 2019. The black horizontal lines (solid, measurements; dashed, model) highlight the average upper mixed layer depth and the depth of minimum temperature. The green horizontal lines highlight the average position of the maximum vertical density gradient. The glider data were averaged in the chosen area (Fig. 1), and the model data in a 1×1 km window (Fig. 1).**





**Figure 4: Dynamics of vertical stratification presented as Brunt-Väisälä frequency squared based on glider data (panels a, c) and model data (panels b, d) in May–August 2018 (a, b) and 2019 (c, d). The glider data show the average profile per section in the chosen area (Fig. 1), and the model data an average profile in a 1×1 km window (Fig. 1).**

To identify whether the changes in the distribution of spice revealed for the glider missions are characteristic of seasonal development, average spice distributions derived from the model data in May–August 2018 and 2019 were analysed. As seen in Fig. 5, the density range between the UML and CIL depths increased from May to August. Like the UML depth, the





depth of the maximum density gradient moved to the lower densities from spring to late summer, except between July and August 2018.

In May, the maximum spice intensity was at lower densities than the UML depth and the depth of the maximum density gradient. In June–August, the maximum spice intensity was identified around the density corresponding to the depth of the maximum density gradient. High spice intensities were characteristic for a relatively broad density range, which increased from June to August, as also the pycnocline covered a larger density range in July–August than in June. The spice maxima were mostly immediately below the maximum density gradient (except June and July 2019). Furthermore, in June 2018 and

August 2019, another local maximum of spice intensity was detected closer to the sea surface (at low densities).

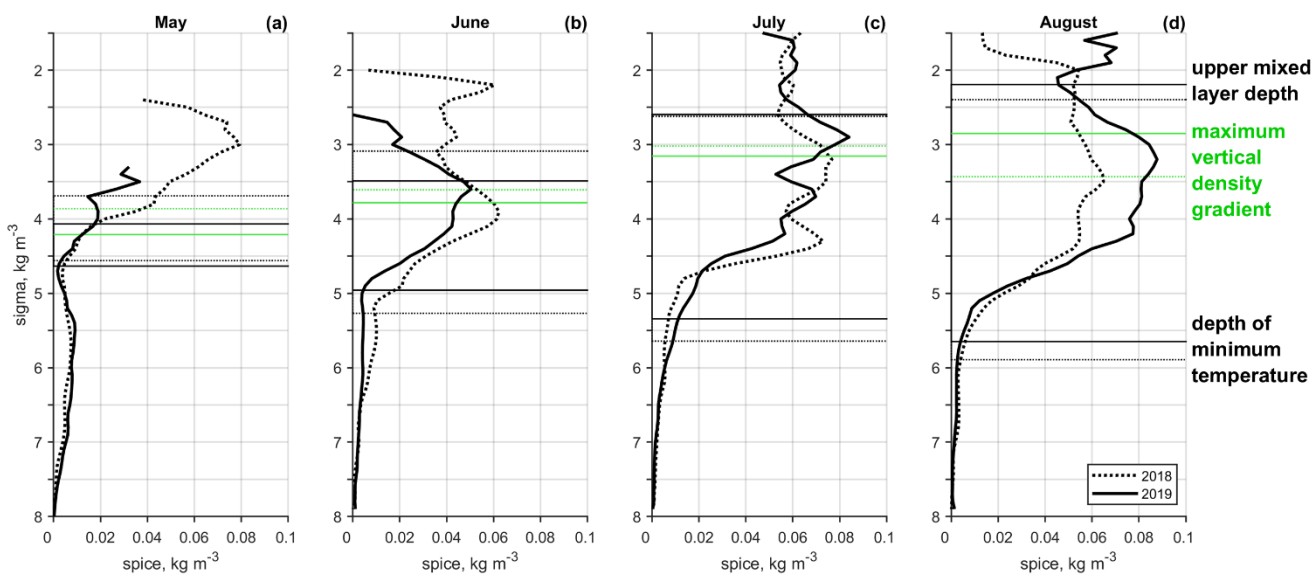

**Figure 5. Monthly standard deviation of spice from May to August in 2018 (dashed) and 2019 (solid). Panels a–d stand for different months. The black horizontal lines highlight the average upper mixed layer depth and the depth of minimum temperature. The green horizontal lines highlight the average position of maximum Brunt-Väisälä frequency (vertical density**
**gradient) referred to as the upper pycnocline (UP). The model data were averaged in a 1×1 km window (Fig. 1).**

### 3.2 Background forcing and large-scale and mesoscale dynamics

Background forcing in the study area in May–August 2018 and 2019 was characterized by mostly positive net surface heat flux (Q) and variable wind conditions (Fig. 6). There were periods with negative Q at the beginning of June and July 2018 and in August 2018, as well as at the beginning of May and July 2019 and late July and early August 2019. Note that easterly

and north-easterly winds could generate upwelling events near the southern coast of the GoF, while westerly and south-westerly could lead to downwelling events. Monthly wind roses suggest that upwelling favourable winds dominated in the study area in July 2018 and were well represented in May 2018 and May and July 2019 (Fig. 6a, c). Downwelling favourable



winds dominated in June of both years. Southerly winds were frequent in August of both years, and northerly winds occurred frequently in July 2019.

Wind speed rarely exceeded 10 m s$^{-1}$, except in June 2018, when three periods with strong westerly winds were observed. These winds leading to vertical mixing and downwelling in the study area caused a deepening of the thermocline and weakening of stratification in the upper layer in June 2018 (Fig. 4b). Similarly, downwelling favourable winds and negative Q at the beginning of July 2019 resulted in the weakening of vertical stratification (Fig. 4d). There were several weekly or bi-weekly periods with calm weather, namely in May and July of both years allowing the strengthening of vertical

stratification. In August, SW–W winds dominated in both years, and Q was no longer predominantly positive.

We suggest that wind forcing and surface heat flux played a crucial role in shaping the thermohaline variability in the area, which differed between the two years. To describe the development of background hydrographic conditions in relation to atmospheric forcing in more detail, we present some characteristic situations from spring–summer 2018–2019.

In May 2018, NE–E winds (Fig. 6b) triggered upwellings along the southern coast, evident in narrow coastal regions with

low temperatures (Fig. 7a). A broader westward flow along the southern coast transported warmer, less salty surface water from the east into the study area (Fig. 7a, d, g). In June 2018, strong SW–W winds led to downwelling along the southern coast (Fig. 7j, m, p). In May 2019, upwelling along the southern coast was more intense than in May 2018 (Fig. 7b, e, h), driven by a strong E wind impulse on May 22–23 (Fig. 6d). By June 2019, upwelling subsided as SW–W winds dominated, creating rotating structures in the gulf's centre (Fig. 7k, n, q). In July 2019, NE–E winds during the last five days of the

month (Fig. 6d) caused upwelling along the southern coast (Fig. 7c, f). A strong outflow from the gulf developed in the surface layer (Fig. 7i). This outflow weakened in early August as northerly winds prevailed and average negative heat fluxes were observed (Fig. 6d). However, a westward flow of fresher water persisted, making the study area a transition zone between different water masses (Fig. 7l, o, r).

Thus, the dynamical background varied substantially during the study period. Depending on wind forcing, upwellings of

varying intensity and associated westward currents (see Fig. 7a–i), downwellings (Fig. 7j, m, p), and eddies (Fig. 7k, n, q and 7l, o, r) developed at the mesoscale. SMS variability in relation to these features on the background of the development of vertical stratification is analysed in the following subsection.





**Figure 6. The daily average net surface heat flux, Q, (red, dashed line showing the monthly average), wind speed (black) and direction (blue) in spring–summer 2018–2019 (Panels b and d). Panels a and c show the wind roses from May to August for both years, respectively.**





**Figure 7. Thermohaline variability in the Gulf of Finland in spring–summer 2018–2019. Panels show the surface (0–3 m) temperature (a–c and j–l), salinity (d–f and m–o), and velocity distributions (g–i and p–r). Presented dates in the first column are 24.05.2018 (a, d, g) and 23.06.2018 (j, m, p), in the second column 24.05.2019 (b, e, h) and 05.06.2019 (k, n, q), and in the third column 31.07.2019 (c, f, i) and 14.08.2019 (l, o, r). The magenta box indicates the 10×10 km study window.**




### 3.3 Submesoscale variability, instabilities and subduction

The intensity of submesoscale variability characterized by the root mean square of spice peaked on several occasions, which partly could be related to concurrent maxima of relative vorticity but often occurred in periods of relatively low Ro values
(Fig. 8). For instance, in 2018, temporal maxima in spice intensity occurred simultaneously with Ro maxima on 25–27 July and at the beginning of August, right after a temporal Ro maximum on 15–16 August and during periods of low Ro on 18 June and 11–13 July (Fig. 8b). Similarly, in 2019, temporal spice maxima coincided with Ro maxima in August, while the largest spice intensity values on 22–30 July were revealed on the background of relatively low Ro values (Fig. 8d).

Note that spice intensities calculated from glider data and model data agree qualitatively well, considering that for the latter,
data from the time series box were used (see the box in Fig. 1 and compare Figs. 8a and 8b and Figs. 8c and 8d). Furthermore, on average, the spice intensity increases from May to August due to the development of seasonal stratification (which is at its maximum in July–August) and enhancement of background gradients of both temperature and salinity. Relative vorticity also undergoes an average increasing trend from May to August, although some strong wind forcing events (as in late June 2018) could create temporal high peaks in Ro (Fig. 8b).

We picked up three occasions when glider and model data were available to demonstrate the relationship between the distribution patterns of parameters indicative of the development of submesoscale instabilities/processes and spice. The intensification of spice in the second half of May 2018 observed by glider measurements (Fig. 8a) and simulated by the model (Fig. 8b) coincided with the formation of horizontal buoyancy gradients and transport of less saline and warmer waters from the east (Fig. 7j, m). First, an increase in Ro (averaged over the time series box) towards O(1) occurred, and
then the spice intensity peaked (Fig. 8b). Figure 9a, d and g demonstrate the presence of structures with O(1) Ro, Ri < 1 and relatively strong horizontal buoyancy gradients on 24 May. Elevated spice intensity was detected near these elongated structures (Fig. 9j).

In late May 2019, an intense upwelling occurred along the southern coast (see an example on 24 May in Fig. 7b, e), but no significant increase in the spice intensity and Ro was observed (Fig. 8d) in the time series box. Ro and slightly the spice
intensity increased at the beginning of June, coinciding with the changing wind forcing (Fig. 6d). A cyclonic eddy (diameter ~10 km) formed in the study area, accompanied by a larger anticyclonic eddy in the offshore area (Fig. 9b). Order one Ro from 5 to 11 June indicated the persistence of this cyclonic feature in the time series box (Fig. 8d). Figure 9b, e, h and k illustrate elevated spice intensity at the periphery of the cyclonic eddy coinciding with narrow stripes of relatively high buoyancy gradients and low Ri.

The high spice intensity in the third trimester of July 2019 (Fig. 8c, d) may be linked to the varying wind forcing and the development of a westward current along the southern coast of the GoF (Fig. 7i). Westerly winds at the beginning of July were followed by upwelling favourable winds on 7–10 July, which evoked the westward coastal current. After that, a period of variable winds occurred, and a new pulse of easterly winds intensified this coastal current. However, a relatively low Ro was detected on the surface in the time series box during this period (Fig. 8d). Nevertheless, as seen in Fig. 9c and f, a region



of high Ro and horizontal buoyancy gradients linked to the intense coastal current were located to the south of the study area. Interestingly, patches of low Ri were also revealed to the north from the region of the highest surface current speeds and patches of high spice intensity were revealed even further to the north (Fig. 9i and l). Since also later, episodic increases in the spice intensity in the subsurface layer (for instance, observed on 8–9 and 13–14 August 2019, Fig. 8c) did not correlate with the peaks in the Ro at the surface, we analyse this situation in more detail.





**Figure 8.** The root mean square spice in the density range limited from the surface to the depth of minimum temperature in spring–summer 2018–2019. Panels a and c show the intensity of spice along the glider trajectory. Panels b and d show the results from the model, which were calculated in a 10×10 km study window. Black line shows the spice and red line the root mean square Rossby number on the surface.







**Figure 9.** The quantities indicative of the SMS regime on the sea surface and the intensity of spice in the density range limited by the depth of minimum temperature in spring–summer 2018–2019. An example date is shown from each glider mission period (I–III stand for May–June 2018, May–June 2019, and July–August 2019, respectively). Panels show Rossby number (a–c), squared horizontal buoyancy gradient modulus (d–f), Richardson number (h–i), and spice intensity (overlapped with surface currents) distributions (j–l), respectively. The magenta box indicates the 10×10 km study window, and the black lines in panel l marks the transects shown in Fig. 10.





**Figure 10. Current fields at 0 m (a; left) and 10 m (a; right, with dashed lines showing the location of sections and the magenta box the 10×10 km study window) depth on 26.07.2019. The colours indicate the speed, and the arrows indicate the direction of currents. Contours are shown with a step of 0.05 m s$^{-1}$. Vertical sections of temperature (b), salinity (c), Ro (d) and density anomaly, and spice (e) along meridians 25.10° E (left column), 25.25° E (middle column), and 25.40° E (right column) on 26.07.2019.**





The distributions of currents, together with vertical sections of temperature, salinity, density, Ro and spice (Fig. 10), allow us to suggest the origin of intense submesoscale variability in the subsurface layer of the study area. By 26 July 2019, a strong coastal current had emerged (Fig. 10a), and high variability in temperature and salinity was detected in the subsurface layer of the three selected sections (Fig. 10b, c). The highest Ro values in the area were associated with the current and topographic features (Fig. 10d), while Ro values were relatively low in the offshore area (see left panel in Fig. 10d and Ro

curve in Fig. 8d). The isopycnals, which were in the subsurface layer offshore, were close to the sea surface at the sites with high Ro. Along-isopycnal spice was high in a much larger area (Fig. 10e) than the intense coastal current at all sections and in the time series box (Fig. 8d). It suggests that the high SMS variability in the offshore area could originate from the mixing of converging water masses with different TS-characteristics likely due to the SMS instabilities of the observed current.

        Distributions of surface currents and spice intensity in a larger area on 29 July 2019 (Fig. 11b, d; it characterizes the same

situation as presented in Fig. 9, right column and Fig. 10) demonstrate a probable link between the coastal current, topography, configuration of the coastline and SMS variability. The spice intensity is the highest along the coast and in offshore areas where the current turns off the coast, especially westward (downflow) from the coastal and topographic irregularities. It can be noted that spice intensity is generally higher west from the largest peninsulas at 25.50–25.75° E and low in the east. Smaller-scale irregularities in temperature and salinity distributions are evident at almost all topographic

features reaching shallower than 40 m (Fig. 11f, h). Also, at longitudes 25.05–25.25° E (it corresponds approximately to the time series box), SMS variability at and below the thermocline and average spice intensity are relatively high (spice intensity was up to 0.2 kg m$^{-3}$; Fig. 11j).

        We compared the described patterns in the case of upwelling with those during downwelling development in early July 2019. In the case of downwelling, the current field was different from that described above, with prevailing shoreward

transport and eddies in the surface layer (Fig. 11a). In contrast to the upwelling situation (Fig. 11d), spice intensity on 3 July 2019 was generally higher east of the larger peninsulas (Fig. 11c). The thermocline (upper pycnocline) was located deeper, and high SMS variability was associated with all topographic features (Fig. 11e, g, i). However, the average level of spice intensity between these topographic features was significantly lower in downwelling than in upwelling. It suggests that topographic features can trigger SMS instabilities and mixing, but SMS patches can travel offshore along slanted isopycnals

only when there is upwelling. Such upwelling-related SMS subduction is directed against the secondary circulation at the mesoscale with offshore flow in the surface layer and onshore flow in the subsurface layer.





**Figure 11. Distributions of surface currents (a, b) and spice intensity (c, d) in a larger area (with dashed line showing the location of section and the magenta box the 10×10 km study window) on 03.07.2019 (left column) and 29.07.2019 (right column). Vertical sections of temperature (e, f) and salinity (g, h), and graph of spice intensity (i, j) are shown along parallel 59.71° N.**



## 4 Discussion

Despite the ubiquity of SMS variability on the sea surface in the Baltic Sea observed via satellite images (e.g., Lavrova et al., 2018) and their recognized implications for energy cascade (Naveira Garabato et al., 2022) and biogeochemistry (Mahadevan, 2016) in other marine systems, the dynamics and impacts of SMS processes are not widely explored in the

Baltic Sea. Both numerical modelling and observational studies confirm their occurrence during the upwelling (Lips et al., 2016a; Väli et al., 2017) and storm events (Carpenter et al., 2020; Chrysagi et al., 2021). Zhurbas et al. (2022) suggested the SMS processes in the UML are separated from those in the layers below. The glider observations analyzed by Salm et al. (2023) confirm the presence of SMS variability in the subsurface layers, suggesting that the high-resolution measurements gathered with autonomous devices like gliders are essential for studying the SMS processes in the Baltic Sea.

The complex physical environment of the Baltic Sea is known to pose a challenge for numerical models, often resulting in biases in the simulated salinity and/or temperature fields, as well as inaccuracies in capturing the pycnoclines correctly (e.g., Gröger et al., 2022; Hordoir et al., 2019; Liblik et al., 2020; Väli et al., 2013). We showed in the present study that the model with the SMS permitting grid spacing could simulate the development of the vertical structure of the water column and the SMS variability reasonably well – the general spice variability exhibiting consistent patterns in both the measurements and

the simulation. Similarly, Chrysagi et al. (2021) successfully compared the GETM simulation and the observations by replicating the structure of an observed cold SMS filament in the model. Studies in oceanic settings have demonstrated the pronounced influence of SMS processes on mixed layer depth (e.g., du Plessis et al., 2017). This suggests that accurate simulation of SMS processes in the models could serve as a vital bridge for predicting the development of vertical stratification, coastal-offshore exchanges, etc.

Spice, which varies from cold and fresh to warm and salty, provides a dynamically passive tracer. We defined spice through along-isopycnal temperature and salinity anomalies with respect to the spatial mean considering typical internal Rossby deformation of 4 km. Some studies have defined spice by analysing the whole dataset (all corresponding spatial scales) and have discussed the tracer spectra (Jaeger et al., 2020; Klymak et al., 2015). We propose that the scale-dependent spice well emphasizes the SMS variability by revealing the patchy structure around the features characterized by the order-of-one

Rossby number and large vertical velocities. Our analysis of spice intensity revealed that tracer patches captured in the vertical structure appear as elongated regions. These dimensions closely match estimations by Zhurbas et al. (2022), who suggested that the SMS structures are approximately 10–20 km in length and 1 km in width. Additionally, our simulation exhibited a small cyclonic vorticity within the study area in spring 2019, with spice following its periphery. Similar results by Yang et al. (2017), who observed heightened mixing in the eddy periphery, are consistent with the concept that the SMS

flows provide a route for energy toward smaller scales.

The SMS tracer variance was consistently observed during all glider missions and throughout the modelling periods from May to August 2018 and 2019. Several observational studies have addressed the seasonal cycle of the SMS flows (e.g., Buckingham et al., 2016; Callies et al., 2015; Thompson et al., 2016). Thompson et al. (2016) demonstrated that the



conditions favourable to the SMS instabilities vary throughout the year, with the SMS flows being more prevalent in the
winter. The seasonality in atmospheric forcing and stratification in the Baltic Sea would also suggest stronger SMS flows in
winter than in spring-summer, as simulated by Väli et al. (2024). Spice intensity, as it is defined in our study, reveals
submesoscale flows only when both temperature and salinity contribute to density changes. For instance, in winter, when the
sea surface temperature varies near the temperature of maximum density, salinity almost alone defines the water density, and
the along-isopycnal spice must be very low. It does not mean that the submesoscale flows do not exist; they do not leave this
signature in winter. It could explain an overall increasing trend in SMS intensities from spring to late summer with the
warming up of the upper layer and development of the seasonal thermocline but not the observed temporal peaks in spice
intensity. We argue that the spatial and temporal peaks in spice intensity indicate the instances with high activity of SMS
processes.

Recent studies have shown that surface heating can suppress SMS flows, and surface cooling leads to more active SMS
flows, but favourable wind forcing can enhance SMS activity also under surface heating (Peng et al., 2021; Shang et al.,
2023). On the other hand, when the wind forcing changes or reduces, SMS frontal instabilities can develop that result in
restratification of the UML, as shown by glider measurements and simulations in the Baltic Sea (Carpenter et al., 2020;
Chrysagi et al., 2021; Salm et al., 2023). Horizontal buoyancy gradients are essential for SMS development (e.g., Bosse et
al., 2021; Ramachandran et al., 2018;), and multiple factors contribute to creating these gradients in the GoF (e.g., Lips et al.,
2016a). The predominantly positive Q fosters water column stratification in spring and early summer, although at different
rates along the coastal and open sea areas (e.g., Lips et al., 2014). The inclination of the thermocline depends on the wind
forcing (Liblik and Lips, 2017), and occurrences of upwellings/downwellings effectively form large horizontal temperature,
salinity and density gradients in spring–summer (e.g., Kikas and Lips, 2016). The SMS processes come into play gaining
energy from created lateral buoyancy gradients (Boccaletti et al., 2007).

We suggest that the submesoscale spice maxima observed above the maximum vertical density gradient in spring and in
conditions of positive Q and low wind forcing were signs of SMS processes acting to restratify the deeper portion of the
UML. The secondary peak of spice intensity near the sea surface in June 2018 and August 2019 (Fig. 5) could be related to a
characteristic forcing pattern from these months – strong wind mixing events were followed by a period of weaker winds and
positive Q. Thus, the submesoscales can act to restratify the UML near the sea surface (positive Q) and at its bottom, due to
lateral buoyancy gradients, as also noted by Miracca-Lage et al. (2024).

Cooling (negative Q) and wind-induced turbulence likely steer the dynamics within the UML, leading to its deepening.
However, the SMS processes can also emerge below the UML and the maximum vertical density gradient, as was evident in
July–August 2019, when both the winds and Q were more variable. Dove et al. (2021) have shown that when the base of the
mixed layer deepened in regions of high eddy kinetic energy, a simultaneous increase in spice and oxygen concentration was
observed below the mixed layer. Our data from July–August 2019 showed that high spice intensity in the subsurface layer
did not coincide with the elevated Ro at the sea surface at the same site. Instead, a baroclinic coastal current, varying in
intensity and shape, was present nearby. We noted high relative vorticity near the topographic features and surfacing of



isopycnals, which were characterized by high spice intensity in the offshore subsurface layer. Similar subduction along the slanted isopycnals associated with the submesoscale activity at the upwelling front has been reported by Hosegood et al.

(2017). Capo et al. (2023) suggested that flow-topography interactions generate vorticities, which are transported offshore in the subsurface layers.

We showed that submesoscale variability in the offshore subsurface layer differs between upwelling- and downwelling-dominated background conditions. The winds favourable for the generation of upwelling (downwelling) in the southern GoF strengthen (weaken) vertical stratification and move the pycnocline upward (downward) towards the coast (Liblik and Lips,

2017). It could be suggested that coastal downwelling favours vertical turbulent mixing due to the weakening of vertical stratification, and SMS features are less visible. Furthermore, if SMS subduction occurs, the patches are transported along the slanted isopycnals shoreward from the downwelling-related baroclinic current. It explains why less intense SMS variability is observed offshore while it is high near the topographic features (Fig. 11 left column).

Ruiz et al. (2019) showed that in a mesoscale front, 80% of the downward vertical flux of phytoplankton can be explained by

the SMS subduction. It agrees with the observation of high phytoplankton biomass patches detected in connection to the submesoscale intrusions below the depth of the strongest vertical density gradient (Lips et al., 2011) and the accumulation of phytoplankton along the depressed isopycnals at the base of anti-cyclonic circulation cells in the GoF (Lips et al., 2010). Väli et al. (2024) revealed that the southern coastal area of the GoF is a region with the highest intensity of relative vorticity in the surface layer at the submesoscale in the Baltic Sea. Thus, the proposed SMS subduction process could feed transport and

mixing in the offshore subsurface layer by exerting energy from unstable coastal currents over varying topography. These results accentuate the substantial role of SMS activity in the surface and subsurface layers of the Baltic Sea and its potential contribution to the vertical and coastal-offshore transport of heat, salt, nutrients, oxygen, and carbon.

**Code Availability Statement**

Scripts to analyse the results are available upon request.

**Data Availability Statement**

The glider data sets analysed for this study can be found in online repository: doi:10.17882/96561 SEANOE. Model data is available upon request.



**Author Contributions**

KS was responsible for processing the glider data, analysing observational and model data and writing of the paper with
contributions from GV, TL, and UL. UL contributed to the analysis setup and supervised the work. UL, TL, and KS
participated in designing the surveys and piloting the glider. GV was responsible for the modelling activities.

**Competing interests**

The authors declare that they have no conflict of interest.

**Acknowledgments**

We would like to thank our colleagues and the crew of RV Salme for their assistance during the deployment and recovery of
the glider. The computational resources from TalTech HPC are gratefully acknowledged. GETM community in Leibniz
Institute for Baltic Sea Research (IOW) is gratefully acknowledged for maintaining and supporting the model code usage
and development.

**Funding**

This work was supported by the Estonian Research Council grant (PRG602). The glider mission in the summer of 2019 was
carried out as part of LAkHsMI (Sensors for LArge scale HydrodynaMic Imaging of ocean floor) project, which received
funding from the European Union's Horizon 2020 Research and Innovation Programme grant No 635568.

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
