# Peer review of "Forcing-dependent submesoscale variability and subduction in a coastal sea area (Gulf of Finland, Baltic Sea)"

_EGUsphere, 2024_

## Referee Comment (RC1)

**Review of**: *"Forcing-dependent submesoscale variability and subduction in the coastal sea area (Gulf of Finland, Baltic Sea)"* by Salm K. et al., 2025

The manuscript presents glider-based hydrographic observations during three missions conducted in the Gulf of Finland in the summers of 2018 and 2019, supported by high-resolution GETM model simulations. The authors aim to investigate the dependence of submesoscale variability on atmospheric forcing and background hydrographic structure, and to explore mechanisms of tracer subduction driven by topographic and baroclinic instabilities. Tracer variance is quantified via the "spice" parameter, which represents thermohaline variability that does not affect density.

The topic is relevant and timely, and the integration of observational and high-resolution modeling approaches is promising. However, the manuscript requires major revisions to meet publication standards and to convincingly support its stated objectives. Below are my detailed comments:

**General Comments**

- **Clarity and Structure**: The manuscript requires significant reorganization and revision for clarity. The writing style is often unclear, with missing or incorrect articles, tenses, and sentence structures throughout.
- The **Introduction** lacks a clear narrative and needs to better articulate the study's motivation, background, and novelty relative to existing work in the GoF.
- **Consistency and Rigor**: A number of concepts, such as "spice", "tracer variance", and various metrics (e.g., Rossby number, buoyancy gradients), are introduced too late, without sufficient explanation or justification. Terminology should be defined clearly and used consistently.
- **Use of Observations vs. Model**: The manuscript leans heavily on the model results, and the glider observations appear to be used mostly for validation. This raises concerns about how well the observational data actually constrain or inform the main findings.
- **Figures and Analysis**: Several figures are difficult to interpret due to inconsistent domains, axes, or color scales. Some key diagnostics are not shown (e.g., vertical velocity, currents (depth vs time)), limiting the reader's ability to assess the conclusions about subduction and mixing.
- A **Conclusion section is missing** and the Discussion lacks direct references to the figures and results.

**Specific Comments:**

*Abstract: the abstract has to be improved so it matches text body and the analysis and it is self-explanatory so the reader knows what exactly was done.*

- L2: Rephrase the second sentence—it's unclear and does not reflect the analysis accurately.
- L4: Specify what "tracer variance" refers to (i.e., spice variance).
- L6: Clarify "around UML" – upper mixed layer? Provide depth range.
- L10–13: Specify what atmospheric forcing is meant; consider merging sentences.
- L14: "likely SMS flows" – was this demonstrated? If so, how?
- L15: Be specific—"high tracer variance" of what exactly?
- L16: The final sentence is vague—was this shown, or is it speculative?

*Introduction (L20–85): this needs to be seriously improved as the writing style is not good, the paragraphs need to be reorganized so the background, motivation and what will be done and why is clear to a reader.*

- L20: Replace "forces" with a clearer term
- L21: Expand the second sentence to explain the background of SMS dynamics.
- L23: Explain why SMS features are important—link to physical or biogeochemical processes.
- L25: Add citations to definitions of SMS.
- L28: Rephrase the final sentence and state the region explicitly.
- L31–42: Revise for English grammar and completeness. "The" and "of" are frequently missing.
- L36–38: Add "semi-enclosed" to describe the Baltic Sea.
- L41: Specify freshwater input sources—mention the main rivers.
- L45: "In contrast to the open ocean" implies salinity is unimportant there—rephrase.
- L46–47: Sentence unclear—needs rewording.
- L48: Clarify whether Lips (2009) and Vali (2017) estimated mixing or just described it.
- L57–58: Replace "captured" and "prevalence" with more precise terms.
- L60–61: What is the key modeling advantage? State it up front.
- L62–64: Introduce the glider earlier; explain why the upper half of the water column is the focus.
- L65: Provide the exact mission durations.
- L66: Rephrase "favored".
- L68: Define "tracer variability" – first mention needs explanation.
- L70–72: Why is only spring-summer analysed? given the model covers a longer period. The two summers you analyze are similar or different?
- L72–75: Clarify the relation to Salm (2018)—same dataset?
- L77: Explain why spice is used and what it captures. This is not a common term.
- L81–85: The hypotheses should be clearly formulated and tested in the Results.

**2. Materials and Methods**

**2.1 Glider Observations**

- L90: Grammar issues—"Three missions were performed…"
- Specify exact dates, transect directions, and water depth coverage.
- How was data quality-controlled? Cite appropriate methods.
- Why were transects oriented differently between missions?
- L98: Describe how the data were interpolated (vertical/horizontal resolution).
- Fig. 1b: Include a broader regional map with coastline for context.

**2.2 Model**

- Clarify whether "adaptive vertical coordinates" refers to sigma or z-coordinates.
- L110: Why was the model vertically interpolated? Was this to match glider data?
- Use consistent tenses (past for methods).

**2.3 Analysis**

- L122: Why use 4 km filtering? What would 2 or 7 km yield? Discuss sensitivity.

- L128–130: Refer to Alenius et al. (2003); discuss how scale selection affects variance.
- L140: Define how spice variance is calculated—add the equation.
- L148: "For it" – unclear, remove.
- L150–155: How was $N^2$ calculated? What vertical spacing and smoothing were used?
- L155- this is a Result or it could be also in the Introduction- move it
- Why the features can be displaced in the model- this should be covered in the introduction
- Parameters such as Rossby number, Ri , buoyancy gradients should be at least mentioned in the Introduction, what the analysis will be performed and why?
- L170: central 'difference' ? 'above' missing words
- Why the wind components were smoothed?
- Fig 2. This should be presented in the separate section e.g model validation or section 3.1 , the spice is shown in density domain, but the rest in time-depth domain, why? What about the currents and vertical velocities? They are important for SMS, maybe not in the observations but the dynamics can be shown in the model.
* * *
**3. Results:**
While the section presents several relevant observations and model outputs, it lacks key quantitative metrics to support the conclusions. If the main findings rely primarily on model-derived interpretations of physical processes, this should be substantiated with appropriate statistics and objective measures. For example, the influence of topographic steering is not sufficiently demonstrated, and vertical velocities—crucial for diagnosing subduction or vertical exchange—are not shown. Including such diagnostics would significantly strengthen the analysis and the credibility of the inferred processes.

**3.1 Validation:**
- Correct the language, there are some missing words e.g in the title etc.

- L181–185: Provide not only mean differences but also standard deviations.
- L190: Define "slight" differences numerically.
- L192–196: Indicate which depths the model fails to resolve secondary maxima.
- L197: If stratification was weaker, state it clearly.
- L205: Are the largest spice values associated with maximum T/S gradients? Clarify.
- Fig. 3: Instead of mean-removed slices, show actual matched cross-sections (as in Fig. 4).
- Fig. 4: Difficult to compare due to gaps in glider profiles. Consider interpolating data.
- L220: This sentence is confusing—aims should be explained in the Introduction.
- L225–235: Clarify whether text refers to model or observations.
- Fig. 5: Clearly label data sources (glider vs. model).

**3.2 Wind Forcing**

- Are wind differences between years sufficient to explain SMS variability?
- Fig. 7: Add mean wind arrows for reference.

**3.3 Submesoscale Indicators**

- L279: 'Maxima of relative vorticity'—show this in a figure.
- L300: "Changing wind forcing" – specify exact changes.
- Fig. 8: Why show minimum temperature instead of UML?
- Relationship between surface Ro and spice variance should be discussed quantitatively. E.g correlations etc.
- Discuss limitations of model—e.g., can EO data be used for validation? In the summer this should be possible
- Fig. 10: Use consistent axes (depth vs. density space); add vertical velocity if available.

- L345: "Probable" – vague. Can this be quantified?
- Fig 11. Why you not use UML or vertical currents? For comparison with spice?
* * *
**4. Discussion**

- L367–374: General background not linked to results—consider trimming.
- L385–395: Was vertical velocity shown anywhere? If not, speculative statements must be softened.
- Discuss how "elongated regions" of spice relate to fronts or subduction processes.
- Consider plotting spice vs. Ro for correlation.
- L400: Discussion on winter SMS processes seems out of place—study only covers summer.
- L402: "SMS only visible with T and S gradients"—is this a limitation of the method?
- L409–411: Clarify whether this is your result or literature-based.
- L423–428: Cite figures for all claimed results.
- L437–443: Strong claims based on limited evidence—can they be supported by broader statistics?
- L444–452: This paragraph appears unrelated to the study and could be removed.
* * *
**Missing Conclusion**

- A summary of key findings and a clear answer to the hypotheses are needed.
- Clarify how glider data contributed—was it only for validation?
- Highlight the study's novelty and limitations clearly.
* * *
**Recommendation: Major Revisions**

The manuscript addresses a relevant topic and includes valuable datasets. However, substantial improvements in structure, clarity, and analysis are needed. The roles of observations and model outputs must be better defined, and the analysis should be aligned more closely with the stated objectives.

---

## Referee Comment (RC2)

**Review of Salm et al, *Forcing-dependent submesoscale variability and subduction in a coastal sea area (Gulf of Finland, Baltic Sea)**

This paper aims to observe and understand variability in submesoscale processes within the Gulf of Finland, using a combination of in-situ observations from a glider platform taken in 2018 and 2019, and model simulations using GETM. They found that the depth at which submesoscale processes occurred at varied between spring and summer, and linked this with changes in stratification and atmospheric forcing. Salm et al also highlight the importance of a coastal current for stimulating submesoscale subduction and explore the connections with topography.

They predominantly use spice as a tracer to identify variability in the water column in both the model and observations but describe other parameters that can be used to identify submesoscale variability (such as horizontal buoyancy gradients) for the model results.

The authors present a valuable dataset with glider observations and model simulations, and this is of importance to publish, especially for further understanding of submesoscales in the Baltic Sea. I feel that the presentation and discussion of the results is lacking in structure and some content, and my recommendation is for major revisions.

**General remarks**

- Throughout, language and grammar need to be double checked. Specifically, the word "the" is often either missing, or unnecessarily added in to sentences.
- The structure and motivation for the paper needs to be clarified. It feels that the glider observations have a minor (only validation) role to play and the bulk of the critical analysis is reliant purely on the model simulations.
- Some of the analysis is very reliant on descriptive or more qualitative assessment of figures, or include reference to parameters not presented. The paper could benefit from identifying some more specific or quantitative metrics (such as those used in several of the papers cited and referred to).

**Specific comments**

*Abstract*
- After all other corrections, I would review the abstract to tighten it up, be more specific when referring to dates, review language such as "suggest", "likely" – can you be more specific? What is the implication and impact from your findings?

*Introduction*
- Throughout the introduction; double check grammar and sentence structures. Especially in Lines 20-40 there are lots of additional (or missing) "the"s which can make it a bit trickier to read through.
- Lines 20-25; I would include the importance and impact of SMS flows here (you talk about the links with carbon, heat etc later on).
- Line 45; "In contrast to the open ocean…" ; this is not necessarily true, salinity has a significant stratifying role in many parts of the oean. Please rephrase.

- Line 57: "captured" – remove this word, it doesn't add to the sentence.
- Line 57: "supports the prevalence" – the number of studies doesn't necessarily mean there are lots of SMS flows/variability, but they do highlight the importance. Maybe rephrase?
- Line 60: this suggests that since observational data are limited, your glider data is of high importance? What are the limitations of relying on the model simulations? Can you include some review of literature/other modelling papers in the Baltic that support the usage of high-resolution models for this study.
- Line 65: you introduce the gliders here; could this be mentioned earlier when talking about SMS observations, are there benefits to specifically using gliders when trying to observe SMS flows?
- Line 65: what are the dates for the missions (what does spring-summer mean?)
- Line 75: I'd be interested to see a bit more introduction of spice here for readers who aren't so familiar. Why use spice over other submesoscale parameters also reported in the papers you cite throughout (e.g. horizontal buoyancy gradients, parameterisations of SMS flows through Ekman and Mixed Layer Eddies)?
- Line 86: you refer to a section after section 4/Discussion, but there is no final section.

*Data and Methods*

- Line 90-94: state explicitly the dates, was it the same transect (if not, why not, how are they different), water depths covered. Maybe a table could be useful for that?
- Line 94: "YOs" is a very specific term, can you find an alternative word to describe the half profile (upcast / downcast)?
- Line 96: Is there a reason to interpolate on time rather than space?
- Figure 1: very hard to see the magenta cross for the wind data
- Line 140: put the equation for spice into a separate line to accentuate it.
- Line 142: citation needed for TEOS-10?
- Line 143: how does the choice of 4km impact results? Did you test with higher/lower scales?
- Line 146: You refer to spice variance throughout the paper, but it is predominantly spice that is plotted in the figures. Is it a qualitative assessment of the figures that results in your analysis of the variance, or do you numerically calculate the variance? If so, can you state that here?
- Line 149/150 (and fig 2): Is it possible to plot spice on the same y-axis? Hard to compare between the panels when T and S are plotted against depth and spice is plotted against sigma.
- Line 155: This reads as a result? Maybe rephrase this to introduce this paragraph.
- Line 161: you calculate several SMS characteristics from the model data. Is it possible to calculate some of these from the glider data too (as is done in papers you cite, such as Thompson, du Plessis et al)
- Line 170: Not sure what you mean by "Central scheme…"
- Line 171: Were these gradients used for N2 as well?
- Line 174: Why do you smooth over 6 hours? Is there a motivation for choosing this timescale?

*Results*

- Line 183: "over which ten profiles were gathered": is that per transect of the glider, so in total you have 10x the number of repeats the glider did?
- By averaging the profiles you lose a lot of the benefits and advantages of the glider data – it would be interesting to see the variability in spice (from glider data) over the transect/mission timeline, and to compare that variability to the seasonal variability that the paper is focused on (i.e. is the shorter timescale comparable to the seasonal?).
- Line 187-193: I struggle to understand what magnitude of difference between model and observations is significant. How much can it impact the spice calculations and the final results? From the figures it is clear that the model does not perfectly replicate the observations, what level is acceptable? (e.g. "slight differences" – what does that. Mean?)
- For quite a few of the figures (4, 8, 9, 10 , 11), it could be beneficial to point towards or highlight features that you discuss in the text (e.g. with a small triangle/arrow/line).
- Line 199: "UP" – if you have enough space, I recommend typing this out in full, "upper pycnocline". Try to remove unnecessary acronyms as much as possible.
- Figure 4: can you interpolate the glider data across time? It is hard to pick up the features you discuss due to the individual profiles.
- Missing more quantified discussion of submesoscale characteristics (e.g. line 245: can you calculate how much the wind changes could impact wind induced SMS flows?). I think this links to the somewhat qualitative or descriptive discussion of the spice tracer; adding quantification where possible will strengthen your conclusions.
- Line 278-285: can you put lines/mark out these events on the figure?
- Line 290: highlight on the figure when these three events occur, help the reader out as much as possible by making it easy to follow your arguments and analysis.
- Line 294: you talk about horizontal buoyancy gradients, but there are none shown in Figures 7 or 8) that you refer to. In addition, you are talking about an event in the second half of May 2018, but you refer to figure 7j,m which shows an event in June.
- From line 335; is this intended to be a new subsection? It is somewhat detached from the previous paragraph (maybe this is just the formatting in the draft version though).
- Paragraph around line 2345: Are there any calculations or analysis that can be done to make this more certain? Or rephrasing of your results (a lot of "probable", "suggests" etc).
- Figure 11: can you colour in the bathymetry in black or similar? It took me a while to spot the features that you refer to in the text. And is there space along the bottom of the figure to add a panel showing the overall topography so we have context of what the rest of the bathymetry is like away from the <40 m peaks?

*Discussion*

- Paragraph 367-374: Feels more like a paragraph for the introduction? Or link it to your discussion/results a lot earlier on.
- Line 391: what do you mean by elongated regions? Vertically or horizontally?
- Lines 396-403: the discussion here is comparing to studies that include winter SMS flows / full annual cycles of submesoscale variability (seasonality). The paper presents itself as looking at seasonal variability, but you only look at spring-summer and not the full seasonal cycle. It seems that this is done to only use the model for periods when the observational data is available to validate it, but this results in a compromise on both of

the data sources: we lose the high resolution variability that could be interesting to look at in the glider data, and also lose the full seasonal cycle that the model could present us. Do you trust the model simulations enough to gain a small insight into other seasons?

- Line 404: Can you give some insight into how much this could impact your conclusions? Is this a significant limitation in using only spice as a tracer for submesoscale flows?

I feel that the paper is missing a final conclusion statement or section, to give a summary of the main results and findings. Also a clear comment on the impact/implications of the study, and any limitations or future aspects to explore.

---

## Author Comment (AC1)

**Review 2 of "Forcing-dependent submesoscale variability and subduction in the coastal sea area (Gulf of Finland, Baltic Sea)" by Salm K. et al., 2025**

This paper aims to observe and understand variability in submesoscale processes within the Gulf of Finland, using a combination of in-situ observations from a glider platform taken in 2018 and 2019, and model simulations using GETM. They found that the depth at which submesoscale processes occurred at varied between spring and summer, and linked this with changes in stratification and atmospheric forcing. Salm et al also highlight the importance of a coastal current for stimulating submesoscale subduction and explore the connections with topography.

They predominantly use spice as a tracer to identify variability in the water column in both the model and observations but describe other parameters that can be used to identify submesoscale variability (such as horizontal buoyancy gradients) for the model results. The authors present a valuable dataset with glider observations and model simulations, and this is of importance to publish, especially for further understanding of submesoscales in the Baltic Sea. I feel that the presentation and discussion of the results is lacking in structure and some content, and my recommendation is for major revisions.

Thank you for the thorough review and comments. To better focus and structure the paper, we have defined the following hypotheses. First, we suggest that SMS variability is modulated by both atmospheric forcing, particularly surface heat flux and wind stress, and the background (larger-scale) hydrographic structures, including mesoscale frontal gradients. Second, we propose that topographically induced instabilities of baroclinic coastal currents create favorable conditions for SMS subduction, enabling offshore and downward transport of tracers.

General remarks

- Throughout, language and grammar need to be double checked. Specifically, the word "the" is often either missing, or unnecessarily added in to sentences.
We have thoroughly revised the manuscript for language and grammar. We have carefully reviewed the use of articles (e.g., "the") and corrected inconsistencies or errors throughout the text.

- The structure and motivation for the paper needs to be clarified. It feels that the glider observations have a minor (only validation) role to play and the bulk of the critical analysis is reliant purely on the model simulations.
We have substantially revised the Introduction to improve the presentation of the motivation and objectives of the study. Since we have published an earlier paper based solely on glider data from 2018, we see this paper as an extension of the study by incorporating two more glider missions and high-resolution model results. We have characterized SMS variability based on glider data in relation to forcing and mesoscale background. The model outputs are used to extrapolate (generalize) the findings over a larger spatial area and temporal extent. We have revised the manuscript to better emphasize the role of glider observations beyond validation. Specifically, we state that glider observations revealed maxima of spice above the maximum vertical density gradient during spring missions and below it during the late summer mission. Also, glider data indicated high spice in the sub-surface layer in the case of forcing conditions favorable for coastal upwelling and formation of a long-coastal baroclinic current. Both of these findings were generalized, and potential mechanisms were suggested using model data from a larger area and extended periods.

- Some of the analysis is very reliant on descriptive or more qualitative assessment of figures, or include reference to parameters not presented. The paper could benefit from identifying some more specific or quantitative metrics (such as those used in several of the papers cited and referred to).

We have added numerical values to clarify several previously qualitative assessments. Phrases that may have been misleading, such as those implying reference to parameters not shown, have been revised to more accurately reflect the intended interpretation of the figures and improve clarity.

We agree that incorporating more specific or quantitative metrics can enhance the analysis. Among the three glider missions, only one included a sufficiently long spatial transect to support meaningful calculation of

horizontal gradients or derived diagnostics regarding background (mesoscale) conditions (partly presented by Salm et al. (2023)). Given these constraints, we used complementary metrics derived from the model, which provides spatially continuous coverage and spans over longer periods.

**Specific comments**

**Abstract**

- After all other corrections, I would review the abstract to tighten it up, be more specific when referring to dates, review language such as "suggest", "likely" – can you be more specific? What is the implication and impact from your findings?
We have reformulated the abstract to stress better the main findings and differentiate what is well founded by data and what can be suggested but must be studied in more detail in the future.

**Introduction**

- Throughout the introduction; double check grammar and sentence structures. Especially in Lines 20-40 there are lots of additional (or missing) "the"s which can make it a bit trickier to read through.
We have thoroughly revised the Introduction.

- Lines 20-25; I would include the importance and impact of SMS flows here (you talk about the links with carbon, heat etc later on).
We have reorganized the opening paragraphs to emphasize the importance and impact of SMS flows from the outset.

- Line 45; "In contrast to the open ocean…" ; this is not necessarily true, salinity has a significant stratifying role in many parts of the oean. Please rephrase.
The sentence was rephrased.

- Line 57: "captured" – remove this word, it doesn't add to the sentence.
The sentence was reworded.

- Line 57: "supports the prevalence" – the number of studies doesn't necessarily mean there are lots of SMS flows/variability, but they do highlight the importance. Maybe rephrase?
The sentence was reworded.

- Line 60: this suggests that since observational data are limited, your glider data is of high importance? What are the limitations of relying on the model simulations? Can you include some review of literature/other modelling papers in the Baltic that support the usage of high-resolution models for this study.
Modelling advantage was stated more clearly. We do not present a literature review but refer to earlier modelling studies of SMS variability and processes in the Baltic (e.g. Väli et al., 2017, 2024; Chrysagi et al., 2021).

- Line 65: you introduce the gliders here; could this be mentioned earlier when talking about SMS observations, are there benefits to specifically using gliders when trying to observe SMS flows?
We have relocated the paragraph on glider studies to the third position and improved it.

- Line 65: what are the dates for the missions (what does spring-summer mean?)
The exact mission durations are included in Methods.

- Line 75: I'd be interested to see a bit more introduction of spice here for readers who aren't so familiar. Why use spice over other submesoscale parameters also reported in the papers you cite throuhout (e.g. horizontal buoyancy gradients, parameterisations of SMS flows through Ekman and Mixed Layer Eddies)?
To improve clarity, the explanation of spice was relocated to immediately follow its initial mention.

- Line 86: you refer to a section after section 4/Discussion, but there is no final section.
We have added a Conclusion section that clearly summarizes the main findings and implications of the study.

**Data and Methods**

- Line 90-94: state explicitly the dates, was it the same transect (if not, why not, how are they different), water depths covered. Maybe a table could be useful for that?

We have included the exact dates of the missions and improved the paragraph. Although originally conducted for different research objectives, three glider missions in the GoF, Baltic Sea, collectively provide a data set for this study. The glider profiled the water column from the surface down to depths of 80–100 meters, depending on the position. While under the surface, the glider started to turn around either 4 m before the surface or 5–6 m before the seafloor.

- Line 94: "YOs" is a very specific term, can you find an alternative word to describe the half profile (upcast / downcast)?

We replaced "YOs" with "up- and downcasts".

- Line 96: Is there a reason to interpolate on time rather than space?

We chose to interpolate the glider-derived parameters in time rather than in space because the glider observations are naturally indexed by time due to their profiling nature. Each measurement corresponds to a specific timestamp rather than a consistent spatial location. However, the choice depends on the purpose. In some cases, spatial interpolation – using distance, latitude, or longitude – can be more beneficial, e.g., when comparing repeated measurements or sections taken at the same location.

- Figure 1: very hard to see the magenta cross for the wind data
It was improved.

- Line 140: put the equation for spice into a separate line to accentuate it.
The equation was moved into a separate line.

- Line 142: citation needed for TEOS-10?
The citation was added.

- Line 143: how does the choice of 4km impact results? Did you test with higher/lower scales?
The choice of a 4 km averaging scale offers a practical balance between resolving SMS structures and suppressing high-frequency noise. Smaller scales may exaggerate variability and obscure persistent features, while larger scales risk smoothing out key SMS signals. Thus, 4 km averaging preserves the essential gradients and anomalies linked to SMS dynamics without compromising interpretability. We examined smaller/larger length scales when considering the scale for the glider data.

To illustrate sensitivity to the choice of horizontal scale, we present an example using the same section shown in Figure 2. Despite variations in horizontal scale, the structure and location of spice anomalies remain consistent across all three estimates, supporting the robustness of the observed frontal features. However, the magnitude of anomalies increases with smoother (larger) scales, likely due to spatial averaging. This is further supported by the depth-resolved standard deviation profiles, which show systematically lower spice variability at 2 km and higher values at 7 km, especially in the upper layers. See the referred figures below.

[Figure]

The figure above shows an example of spice distribution using horizontal averaging scales of 2 km (a), 4 km (b), and 7 km (c). Each panel displays the same section, overlaid with density contours at 0.2 kg m⁻³ intervals. The data are based on glider observations from 24–25 May 2018.

[Figure]

The figure above shows the standard deviation of spice calculated from the glider mission conducted between 9 May and 6 June 2018. The colours indicate the horizontal averaging scales used for spice: 2 km (blue), 4 km (black), and 7 km (red).

- Line 146: You refer to spice variance throughout the paper, but it is predominantly spice that is plotted in the figures. Is it a qualitative assessment of the figures that results in your analysis of the variance, or do you numerically calculate the variance? If so, can you state that here?
Spice variance is not calculated in this study. The phrase was reworded.

- Line 149/150 (and fig 2): Is it possible to plot spice on the same y-axis? Hard to compare between the panels when T and S are plotted against depth and spice is plotted against sigma.
We have harmonized axes across related plots where relevant.

- Line 155: This reads as a result? Maybe rephrase this to introduce this paragraph.
We agree. It was moved to Results.

- Line 161: you calculate several SMS characteristics from the model data. Is it possible to calculate some of these from the glider data too (as is done in papers you cite, such as Thompson, du Plessis et al)
Yes, most of the parameters are, in principle, possible to estimate from glider sections (with certain limitations). We did so in an earlier paper (Salm et al., 2023). In this study, we preferred to calculate them (except for spice and characteristics of vertical stratification) from model data, which provide spatially continuous coverage and span longer periods.

- Line 170: Not sure what you mean by "Central scheme..."
The centred finite-difference scheme estimates spatial derivatives by combining forward and backward differences, which correspond to differences taken with respect to neighbouring points ahead and behind in space.

- Line 171: Were these gradients used for N2 as well?
Vertical buoyancy gradient was calculated using a 2 m vertical interval. It is now clarified in the text.

- Line 174: Why do you smooth over 6 hours? Is there a motivation for choosing this timescale?
The wind components in the analysis were smoothed by a Gaussian low-pass filter for 6 h to reduce high-frequency noise and highlight relevant forcing scales.

Results

- Line 183: "over which ten profiles were gathered": is that per transect of the glider, so in total you have 10x the number of repeats the glider did?

We have rephrased the text for clarity. This common area corresponds to approximately 1 km of glider track per section. Therefore, a profile is obtained per section, and the total number corresponds to the number of repeated sections.

- By averaging the profiles you lose a lot of the benefits and advantages of the glider data
 We now present temperature and salinity as we previously did for the squared Brent-Väisälä frequency. See the figure below.

[Figure]

The figure above shows temperature variability based on glider data (panels a, c) and model output (panels b, d) for May–August 2018 (a, b) and 2019 (c, d). The glider data represent average profiles for each section within the selected area, forming a composite data field. The model data correspond to average profiles within a 1×1 km window, providing a profile at each model timestep. The blue and white lines show the UML and CIL depth, respectively.

[Figure]

The figure above shows salinity variability based on glider data (panels a, c) and model output (panels b, d) for May–August 2018 (a, b) and 2019 (c, d). The glider data represent average profiles for each section within the selected area, forming a composite data field. The model data correspond to average profiles within a 1×1 km window, providing a profile at each model timestep. The blue and white lines show the UML and CIL depth, respectively.

– it would be interesting to see the variability in spice (from glider data) over the transect/mission timeline, and to compare that variability to the seasonal variability that the paper is focused on (i.e. is the shorter timescale comparable to the seasonal?).

- Line 187-193: I struggle to understand what magnitude of difference between model and observations is significant. How much can it impact the spice calculations and the final results? From the figures it is clear that the model does not perfectly replicate the observations, what level is acceptable? (e.g. "slight differences" – what does that. Mean?)
The paragraph was revised and the term "slight" has been replaced with specific numerical differences to improve clarity and avoid ambiguity.

- For quite a few of the figures (4, 8, 9, 10 , 11), it could be beneficial to point towards or highlight features that you discuss in the text (e.g. with a small triangle/arrow/line).
We agree. We have marked referred periods and features to improve the readability of the figures (and text).

- Line 199: "UP" – if you have enough space, I recommend typing this out in full, "upper pycnocline". Try to remove unnecessary acronyms as much as possible.
We removed this acronym.

- Figure 4: can you interpolate the glider data across time? It is hard to pick up the features you discuss due to the individual profiles.
The gaps have been removed and discussed periods/features marked.

- Missing more quantified discussion of submesoscale characteristics (e.g. line 245: can you calculate how much the wind changes could impact wind induced SMS flows?). I think this links to the somewhat qualitative or descriptive discussion of the spice tracer; adding quantification where possible will strengthen your conclusions.

- Line 278-285: can you put lines/mark out these events on the figure?
Yes, we did so in the revised manuscript.

- Line 290: highlight on the figure when these three events occur, help the reader out as much as possible by making it easy to follow your arguments and analysis.
Done.

- Line 294: you talk about horizontal buoyancy gradients, but there are none shown in Figures 7 or 8) that you refer to. In addition, you are talking about an event in the second half of May 2018, but you refer to figure 7j,m which shows an event in June.
We checked the text and cited figures. There was a mistake, and we corrected it.

- From line 335; is this intended to be a new subsection? It is somewhat detached from the previous paragraph (maybe this is just the formatting in the draft version though).
We ensured that all the text for subsection 3.3 appears before the figures.

- Paragraph around line 345: Are there any calculations or analysis that can be done to make this more certain? Or rephrasing of your results (a lot of "probable", "suggests" etc).
We have made an additional thorough analysis of the selected situation with high spice, which was detected offshore from the coastal baroclinic current. We explained high spice by the presence of the coastal current and its instabilities. It is evident that such situations emerge when upwelling-favorable conditions prevail and the coastal baroclinic current develops. The conclusions will remain as suggestions (not quantitatively approved) since it is difficult to identify a single reason for instabilities and the observed high spice immediately below the strongest vertical buoyancy gradient. We added/refined figures. We clarified the text in the Results and Discussion section accordingly.

- Figure 11: can you colour in the bathymetry in black or similar? It took me a while to spot the features that you refer to in the text. And is there space along the bottom of the figure to add a panel showing the overall topography so we have context of what the rest of the bathymetry is like away from the <40 m peaks?

The figure was redrawn to also show current vectors in the subsurface layer, and the topography was added as a layer on it.

Discussion

- Paragraph 367-374: Feels more like a paragraph for the introduction? Or link it to your discussion/results a lot earlier on.

We omit general statements and link the discussion to the results of the present study.

- Line 391: what do you mean by elongated regions? Vertically or horizontally?

They are elongated horizontally. We added a reference to the figure in the Results section.

- Lines 396-403: the discussion here is comparing to studies that include winter SMS flows / full annual cycles of submesoscale variability (seasonality). The paper presents itself as looking at seasonal variability, but you only look at spring-summer and not the full seasonal cycle. It seems that this is done to only use the model for periods when the observational data is available to validate it, but this results in a compromise on both of the data sources: we lose the high resolution variability that could be interesting to look at in the glider data, and also lose the full seasonal cycle that the model could present us. Do you trust the model simulations enough to gain a small insight into other seasons?

We do not discuss winter conditions in the revised manuscript.

- Line 404: Can you give some insight into how much this could impact your conclusions? Is this a significant limitation in using only spice as a tracer for submesoscale flows?

It is stated now explicitly that the analysis based on spice (as it is defined in the present study) is best applicable during seasons when both temperature and salinity have comparable contributions to density variations. It is not the case in brackish waters at low temperatures when salinity mostly defines the density variations.

I feel that the paper is missing a final conclusion statement or section, to give a summary of the main results and findings. Also a clear comment on the impact/implications of the study, and any limitations or future aspects to explore.

We formulated concluding remarks as follows: This study demonstrates that submesoscale variability in the Gulf of Finland is strongly modulated by both atmospheric forcing – particularly surface heat flux and wind stress – and background hydrographic structures such as mesoscale frontal gradients. Glider observations, supported by high-resolution modelling, revealed consistent spatial patterns of SMS activity, with spice anomalies concentrated near the UML base in spring and within the thermocline in late summer, demonstrating the vertical sensitivity of SMS features to seasonal stratification. While seasonal stratification played a key role in shaping SMS structure, wind forcing became dominant under weaker surface buoyancy input. High spice variability and subduction signatures were consistently found on the offshore side of a baroclinic coastal current, where sloped isopycnals aligned with velocity and spice gradients indicated downward and lateral transport of surface-layer water masses. The integration of observations and model output allowed for extrapolation beyond individual glider transects, confirming that SMS processes in this coastal sea are both dynamically active and responsive to variations in external forcing. Together, these results clarify the physical mechanisms driving SMS variability and subduction in stratified coastal environments.

References

Chrysagi, E., Umlauf, L., Holtermann, P., Klingbeil, K., and Burchard, H.: High-resolution simulations of submesoscale processes in the Baltic Sea: The role of storm events, J. Geophys. Res.: Oceans, 126(3), doi:10.1029/2020JC016411, 2021.

Väli, G., Meier, H.E.M., Liblik, T., Radtke, H., Klingbeil, K., Gräwe, U., and Lips, U.: Submesoscale processes in the surface layer of the central Baltic Sea: A high-resolution modelling study, Oceanologia, 66 (1), 78−90, doi: 10.1016/j.oceano.2023.11.002, 2024.

Väli, G., Zhurbas, V., Lips, U., and Laanemets, J.: Submesoscale structures related to upwelling events in the Gulf of Finland, Baltic Sea (numerical experiments), J. Mar. Syst., 171, 31–42, doi:10.1016/j.jmarsys.2016.06.010, 2017.

---

## Author Comment (AC2)

**Review 1 of "Forcing-dependent submesoscale variability and subduction in the coastal sea area (Gulf of Finland, Baltic Sea)" by Salm K. et al., 2025**

General Comments

• Clarity and Structure: The manuscript requires significant reorganization and revision for clarity. The writing style is often unclear, with missing or incorrect articles, tenses, and sentence structures throughout.
Thank you for the thorough review. We have carefully revised the manuscript to improve grammar, clarity, and overall flow.

• The Introduction lacks a clear narrative and needs to better articulate the study's motivation, background, and novelty relative to existing work in the GoF.
We have substantially revised the Introduction to clarify the narrative and strengthen the study's motivation and context. The updated text provides a clearer explanation of the relevance of submesoscale processes and the use of spice to characterise them in stratified estuarine systems, such as the Gulf of Finland. We also highlight one aspect of the novelty of our study in terms of combining glider-based observations with a regional high-resolution model to investigate the vertical structure and evolution of spice anomalies, which has not been previously addressed in the GoF context or similar basins.

• Consistency and Rigor: A number of concepts, such as "spice", "tracer variance", and various metrics (e.g., Rossby number, buoyancy gradients), are introduced too late, without sufficient explanation or justification. Terminology should be defined clearly and used consistently.
We have revised the manuscript to introduce key physical and dynamical concepts, including spice, more clearly and earlier. Definitions are now provided at their first mention in both the Introduction and Methods sections, with consistent terminology used throughout.

• Use of Observations vs. Model: The manuscript leans heavily on the model results, and the glider observations appear to be used mostly for validation. This raises concerns about how well the observational data actually constrain or inform the main findings.
We have published an earlier paper based solely on glider data from 2018. We incorporated two more glider missions from the study area to characterize SMS variability in relation to forcing and mesoscale background and used model data to extrapolate (generalize) the findings over a larger spatial area and temporal extent. While the numerical model provides the spatial and temporal coverage necessary to explain the development of submesoscale processes, we have revised the manuscript to better emphasize the role of glider observations beyond validation. Specifically, we state that glider observations revealed maxima of spice above the maximum vertical density gradient during spring missions and below it during the late summer mission. Also, glider data indicated high spice in the sub-surface layer in the case of forcing conditions favorable for coastal upwelling and formation of a long-coastal baroclinic current. Both of these findings were generalized, and potential mechanisms were suggested using model data from a larger area and extended periods.

• Figures and Analysis: Several figures are difficult to interpret due to inconsistent domains, axes, or color scales. Some key diagnostics are not shown (e.g., vertical velocity, currents (depth vs time)), limiting the reader's ability to assess the conclusions about subduction and mixing.
We have reworked several figures to improve clarity and comparability – for example, by harmonizing axes across related plots and adding subplots to include vertical velocity and horizontal currents. New figures were introduced where necessary due to the large number of subplots. These enhancements allow for a more transparent evaluation of subduction and help readers more effectively assess the physical mechanisms discussed in the manuscript.

• A Conclusion section is missing and the Discussion lacks direct references to the figures and results.
We have added a Conclusion section that clearly summarizes the main findings and implications of the study. The Discussion section has been reorganized to directly reference relevant figures and results and includes a more focused comparison with previous studies. These changes improve the cohesion of the manuscript and highlight how our findings contribute to understanding submesoscale dynamics and vertical structure in the GoF.

Specific Comments:

Abstract: the abstract has to be improved so it matches text body and the analysis and it is self-explanatory so the reader knows what exactly was done.

• L2: Rephrase the second sentence—it's unclear and does not reflect the analysis accurately.
• L4: Specify what "tracer variance" refers to (i.e., spice variance).
• L6: Clarify "around UML" – upper mixed layer? Provide depth range.
• L10–13: Specify what atmospheric forcing is meant; consider merging sentences.
• L14: "likely SMS flows" – was this demonstrated? If so, how?
• L15: Be specific—"high tracer variance" of what exactly?
• L16: The final sentence is vague—was this shown, or is it speculative?

Thank you for your suggestions. We have reformulated the abstract to stress better what was done and what are the main findings.

Introduction (L20–85): this needs to be seriously improved as the writing style is not good, the paragraphs need to be reorganized so the background, motivation and what will be done and why is clear to a reader.

• L20: Replace "forces" with a clearer term
The sentence was made more precise.

• L21: Expand the second sentence to explain the background of SMS dynamics.
The sentence was revised, including the SMS intermediate horizontal scale.

• L23: Explain why SMS features are important—link to physical or biogeochemical processes.
We have reorganized the opening paragraphs to emphasize the importance and impact of SMS flows from the outset.

• L25: Add citations to definitions of SMS.
Added.

• L28: Rephrase the final sentence and state the region explicitly.
We have relocated the paragraph on glider studies to the third position, where it now naturally leads into the introduction of the study region.

• L31–42: Revise for English grammar and completeness. "The" and "of" are frequently missing.
Revisions were made to the regional description paragraphs to improve coherence and readability.

• L36–38: Add "semi-enclosed" to describe the Baltic Sea.
Added.

• L41: Specify freshwater input sources—mention the main rivers.
We now mention the Neva River, the largest freshwater source and most relevant in the context of the GoF. We believe listing all major rivers is not necessary in this context.

• L45: "In contrast to the open ocean" implies salinity is unimportant there—rephrase.
The sentence was revised and moved.

• L46–47: Sentence unclear—needs rewording.
The sentence was reworded.

• L48: Clarify whether Lips (2009) and Vali (2017) estimated mixing or just described it.
The sentence was revised for clarity. These works address different aspects of the topic. While Lips et al. (2009) emphasized the role of coastal upwelling in enhancing vertical mixing and nutrient transport, Väli et al. (2017) focused on the SMS structures that arise during such events. Lips et al. (2009) estimated via TS-analysis that the upwelled water comprised approximately 85% intermediate layer water and 15% surface mixed layer water.

• L57–58: Replace "captured" and "prevalence" with more precise terms.
The sentence was reworded.

• L60–61: What is the key modeling advantage? State it up front.

Modelling advantage was stated more clearly (including, models help to extend the findings over space and time).

• L62–64: Introduce the glider earlier; explain why the upper half of the water column is the focus.
We have relocated the paragraph on glider studies to the third position. We focus on the upper half of the water column, where SMS activity is most prominent, and the glider data are more densely sampled, allowing more robust analysis.

• L65: Provide the exact mission durations.
The exact mission durations were included in Methods.

• L66: Rephrase "favored".
The sentence was revised ("associated").

• L68: Define "tracer variability" – first mention needs explanation.
To improve clarity, the explanation of spice was relocated to immediately follow its initial mention.

• L70–72: Why is only spring-summer analysed? given the model covers a longer period. The two summers you analyze are similar or different?
Our study is limited to the spring–summer period, when seasonal stratification develops and becomes well established. SMS processes are known to exhibit seasonal variability (e.g., Wang et al., 2018; Yu et al., 2021), and this time window allows us to isolate conditions, such as upwelling, frontal activity, and spice gradients, when SMS dynamics is most active and relevant to our objective of understanding SMS generation under stratified, wind-driven conditions. Furthermore, spice, as defined in the present study, is best applicable during seasons when both temperature and salinity have comparable contributions to density variations. It is not the case in brackish waters at low temperatures when salinity mostly defines the density variations. For instance, the temperature of maximum density in the Gulf of Finland is about 2.5 °C.

Although the model simulations cover a longer time period, our analysis remains focused on spring–summer to ensure consistency with the observational data and to isolate seasonal conditions favourable for SMS generation. Notably, while both summers analysed share common features – such as surface stratification and wind-driven upwelling events – there are also interannual differences in the timing and intensity of these processes, which are examined in the context of their influence on SMS variability in the Discussion section.

• L72–75: Clarify the relation to Salm (2018)—same dataset?
The explanation was added in the text. This study builds on earlier observations of SMS features in the GoF (Salm et al., 2023), extending the analysis to include all three missions conducted in the same area.

• L77: Explain why spice is used and what it captures. This is not a common term.
The explanation of spice and the motivation for using it was expanded. Among other arguments, spice reveals anomalies and spatial gradients along isopycnals, where SMS processes often act. It serves as a proxy for SMS intensity capturing variability without being masked by vertical excursions of isopycnal surfaces, i.e., internal waves.

• L81–85: The hypotheses should be clearly formulated and tested in the Results.
We refined the hypotheses to enhance their clarity and alignment with the study objectives. First, we suggest that SMS variability is modulated by both atmospheric forcing, particularly surface heat flux and wind stress, and the background (larger-scale) hydrographic structures, including mesoscale frontal gradients. Second, we propose that topographically induced instabilities of baroclinic coastal currents create favorable conditions for SMS subduction, enabling offshore and downward transport of tracers.

2. Materials and Methods

2.1 Glider Observations

• L90: Grammar issues—"Three missions were performed…"
The beginning of this section was reworded, including the exact dates of the missions.

• Specify exact dates, transect directions, and water depth coverage.

These transects were oriented across the southern coast of the GoF, capturing cross-shore variability. The glider profiled the water column from the surface down to depths of 80–100 meters, depending on the position. While under the surface, the glider started to turn around either 4 m before the surface or 5–6 m before the seafloor.

• How was data quality-controlled? Cite appropriate methods.
The raw data were quality controlled following procedures adapted from Argo quality control protocols (Wong et al., 2025).

• Why were transects oriented differently between missions?
Although originally conducted for different research objectives, three glider missions in the GoF, Baltic Sea (Fig. 1) collectively provide a data set for this study.

• L98: Describe how the data were interpolated (vertical/horizontal resolution).
Explanations added. The interpolated data fields had a vertical resolution of 0.5 dbar and a horizontal resolution of 10 min. Interpolation was performed on a two-dimensional dataset using pressure as the vertical coordinate and time as the horizontal coordinate.

• Fig. 1b: Include a broader regional map with coastline for context.
The map is slightly extended.

**2.2 Model**

• Clarify whether "adaptive vertical coordinates" refers to sigma or z-coordinates.
We have specified the description of adaptive vertical coordinates. Such a grid is a generalization of sigma layers with the potential to enhance vertical resolution near boundaries and in layers with strong stratification and shear (Klingbeil et al., 2018).

• L110: Why was the model vertically interpolated? Was this to match glider data?
It is not clear which line/sentence is referred to here. In this section, interpolation is discussed in relation to generating boundary data for the high-resolution model from the coarse-resolution model. The output of the coarse-resolution model at boundaries was vertically interpolated to fixed z-levels for input to a high-resolution model, as there are spatial differences between these models.

• Use consistent tenses (past for methods).
The tenses were checked.

**2.3 Analysis**

• L122: Why use 4 km filtering? What would 2 or 7 km yield? Discuss sensitivity.
• L128–130: Refer to Alenius et al. (2003); discuss how scale selection affects variance.
The chosen length scale is consistent with the internal Rossby deformation radius in the GoF, which is typically 2–4 km (Alenius et al., 2003). The choice of a 4 km averaging scale offers a practical balance between resolving SMS structures and suppressing high-frequency noise. Smaller scales may exaggerate variability and obscure persistent features, while larger scales risk smoothing out key SMS signals. Thus, 4 km averaging preserves the essential gradients and anomalies linked to SMS dynamics without compromising interpretability.

To illustrate sensitivity to the choice of horizontal scale, we present an example using the same section shown in Figure 2. Despite variations in horizontal scale, the structure and location of spice anomalies remain consistent across all three estimates, supporting the robustness of the observed frontal features. However, the magnitude of anomalies increases with smoother (larger) scales, likely due to spatial averaging. This is further supported by the depth-resolved standard deviation profiles, which show systematically lower spice variability at 2 km and higher values at 7 km, especially in the upper layers. See the referred figures below.

[Figure]

The figure above shows an example of spice distribution using horizontal averaging scales of 2 km (a), 4 km (b), and 7 km (c). Each panel displays the same section, overlaid with density contours at 0.2 kg m⁻³ intervals. The data are based on glider observations from 24–25 May 2018.

[Figure]

The figure above shows the standard deviation of spice calculated from the glider mission conducted between 9 May and 6 June 2018. The colours indicate the horizontal averaging scales used for spice: 2 km (blue), 4 km (black), and 7 km (red).

• L140: Define how spice variance is calculated—add the equation.
Upon review, we realized that the sentence regarding spice variance was inaccurate and, therefore, removed. Spice variance is not calculated in this study.

• L148: "For it" – unclear, remove.
Removed.

• L150–155: How was N² calculated? What vertical spacing and smoothing were used?
Vertical buoyancy gradient was calculated using a 2 m vertical interval.

• L155- this is a Result or it could be also in the Introduction- move it
We agree. It was moved to Results.

• Why the features can be displaced in the model- this should be covered in the introduction
This content has been repositioned and is now covered in the Introduction.

• Parameters such as Rossby number, Ri , buoyancy gradients should be at least mentioned in the Introduction, what the analysis will be performed and why?
Parameters and their relevance are mentioned in the Introduction.

• L170: central 'difference' ? 'above' missing words
Centred finite-difference scheme was meant here.

• Why the wind components were smoothed?
The wind components in the analysis were smoothed by a Gaussian low-pass filter for 6 h to reduce high-frequency noise and highlight relevant forcing scales.

• Fig 2. This should be presented in the separate section e.g model validation or section 3.1 , the spice is shown in density domain, but the rest in time-depth domain, why? What about the currents and vertical velocities? They are important for SMS, maybe not in the observations but the dynamics can be shown in the model.
Spice is derived in density space, where isopycnal-referenced anomalies yield the most meaningful physical insights. We ensure that spice captures the variability relevant for detecting SMS dynamics, even if it means sacrificing easy visualization in fixed-depth space. However, to make the presentation clearer, we revised the figures where relevant to display spice in depth coordinates. We also now show vertical velocities where relevant, but state that vertical velocities in the frequency domain of interest are largely contaminated by the vertical movements of isopycnals (internal waves). Thus, their more detailed presentation is not relevant here.

3. Results: While the section presents several relevant observations and model outputs, it lacks key quantitative metrics to support the conclusions. If the main findings rely primarily on model-derived interpretations of physical processes, this should be substantiated with appropriate statistics and objective measures. For example, the influence of topographic steering is not sufficiently demonstrated, and vertical velocities—crucial for diagnosing subduction or vertical exchange—are not shown. Including such diagnostics would significantly strengthen the analysis and the credibility of the inferred processes.

3.1 Validation:

• Correct the language, there are some missing words e.g in the title etc.
We have carefully reviewed and revised the manuscript to correct grammatical and typographical issues, including missing words.

• L181–185: Provide not only mean differences but also standard deviations.
We have updated the text about the UML, maximum vertical density gradient, and CIL depths, providing both mean values and standard deviations when comparing observations and simulations. This offers a more comprehensive representation of variability and model performance.

• L190: Define "slight" differences numerically.
The term "slight" has been replaced with specific numerical differences to improve clarity and avoid ambiguity.

• L192–196: Indicate which depths the model fails to resolve secondary maxima.
We now explicitly indicate the depths at which the model fails to capture the observed secondary maxima in the vertical density gradient (observed at depths of 19-20 m), allowing readers to better assess the limitations of the model.

• L197: If stratification was weaker, state it clearly.
The revised sentence now clearly states that stratification was weaker.

• L205: Are the largest spice values associated with maximum T/S gradients? Clarify.
Yes, largest spice values are close to maximum T/S gradients, but above the pycnocline during spring missions and below it during the late summer mission.

• Fig. 3: Instead of mean-removed slices, show actual matched cross-sections (as in Fig. 4).
• Fig. 4: Difficult to compare due to gaps in glider profiles. Consider interpolating data.
Both figures are revised. An example of sections is shown below (new Fig. 3), where glider data are interpolated to match the model output (although there is no complete coincidence of the data in space from the model and observations). These will be shown instead of average profiles from the missions).

[Figure]

The figure above shows temperature variability based on glider data (panels a, c) and model output (panels b, d) for May–August 2018 (a, b) and 2019 (c, d). The glider data represent average profiles for each section within the selected area, forming a composite data field. The model data correspond to average profiles within a 1×1 km window, providing a profile at each model timestep. The blue and white lines show the UML and CIL depth, respectively.

[Figure]

The figure above shows salinity variability based on glider data (panels a, c) and model output (panels b, d) for May–August 2018 (a, b) and 2019 (c, d). The glider data represent average profiles for each section within the selected area, forming a composite data field. The model data correspond to average profiles within a 1×1 km window, providing a profile at each model timestep. The blue and white lines show the UML and CIL depth, respectively.

• L220: This sentence is confusing—aims should be explained in the Introduction.
The beginning of the paragraph was revised.

• L225–235: Clarify whether text refers to model or observations.
We have clarified in the text that the analysis in this section is based on model-derived spice fields. The paragraph describes seasonal variability using standard deviations of modelled spice.

• Fig. 5: Clearly label data sources (glider vs. model).
This figure shows the standard deviations of modelled spice, which is now explicitly noted in the figure caption.

3.2 Wind Forcing

• Are wind differences between years sufficient to explain SMS variability?
We used both wind data and heat flux estimates to explain temporal variability, including differences between the years. Wind is most crucial for the development of mesoscale features (e.g. baroclinic current), while heat flux may define the potential of SMS activity at the base of the upper mixed layer (if it is large enough compared to the wind mixing).

• Fig. 7: Add mean wind arrows for reference.
We prefer to keep this information in the text since it is not obvious which period should be used to derive a mean vector that is relevant for each situation.

3.3 Submesoscale Indicators

• L279: 'Maxima of relative vorticity'—show this in a figure.
This phrase was potentially misleading and has been revised for clarity to more accurately reflect the intended meaning to describe Fig. 8.

• L300: "Changing wind forcing" – specify exact changes.
We have revised the sentence to specify the observed transition in wind direction.

• Fig. 8: Why show minimum temperature instead of UML?
The depth of the minimum temperature was chosen as a boundary in Fig. 8 because it effectively captures the upper part of the water column, encompassing both the UML and the thermocline. The UML is typically only a few meters thick, and a significant portion of the spice variability is observed not just within the UML but within the thermocline. Therefore, using the minimum temperature depth provides a more representative boundary for analysing upper-layer variability.

• Relationship between surface Ro and spice variance should be discussed quantitatively. E.g correlations etc.
We chose not to emphasize a direct correlation analysis between surface Ro and spice variance. As shown in Fig. 8, there are situations with coinciding high Ro and high spice, high Ro and low spice, and low Ro and high spice. We chose the case with the highest spice (but low surface Ro) and tried to explain the mechanism of its creation.

• Discuss limitations of model—e.g., can EO data be used for validation? In the summer this should be possible
It is not the topic of the Results section and is discussed in the Discussion section with reference to other studies.

• Fig. 10: Use consistent axes (depth vs. density space); add vertical velocity if available.
We have revised Fig. 10 to use consistent axes across all subplots and included vertical and horizontal velocity fields.

• L345: "Probable" – vague. Can this be quantified?
It is difficult to quantify it. We explained this probable link between the coastal current, topography, configuration of the coastline and SMS variability in a descriptive way. In the revised manuscript, an analysis is

added to incorporate wind forcing into this topic in more detail. The conclusions will remain as suggestions (not quantitatively approved). We clarified the text in the Results and Discussion section accordingly.

• Fig 11. Why you not use UML or vertical currents? For comparison with spice?
Thank you. We stress more clearly that high spice below the maximum vertical density gradient occurs when the upper mixed layer is shallower (upwelling-favourable winds prevail). See Fig. 11. As stated above, vertical currents are contaminated by vertical movements of isopycnals (internal waves) and are difficult to directly link with SMS activity.

4. Discussion

• L367–374: General background not linked to results—consider trimming.
We omit general statements and link the discussion to the results of the present study.

• L385–395: Was vertical velocity shown anywhere? If not, speculative statements must be softened.
It is a referred statement. We did not deal with vertical velocities as indicators of SMS activity. The sentences are reworded accordingly to clarify it.

• Discuss how "elongated regions" of spice relate to fronts or subduction processes.
Thank you. We added relevant discussion when analyzing the appearance of high spice in relation to baroclinic currents, their instability and the reasons behind (topography and/or changes in wind forcing).

• Consider plotting spice vs. Ro for correlation.
As we explained above, we do not consider it worthwhile to show (there are situations when the correlation exists and when it does not).

• L400: Discussion on winter SMS processes seems out of place—study only covers summer.
We left it out.

• L402: "SMS only visible with T and S gradients"—is this a limitation of the method?
Yes, it is discussed as a limitation of the method. It is mentioned now explicitly.

• L409–411: Clarify whether this is your result or literature-based.
It is a literature-based statement. We link it with the results of the present study, indicating that our results reveal an opposite tendency (as do some other studies) towards enhanced SMS activity at the base of the UML when surface heating prevails over wind mixing.

• L423–428: Cite figures for all claimed results.
We checked it and cited the figures where relevant.

• L437–443: Strong claims based on limited evidence—can they be supported by broader statistics?
As we explained, adding statistics is challenging, and we base our suggestions on the analysis of specific situations. We agree that the statements need to be softened (as we did in the revised text).

• L444–452: This paragraph appears unrelated to the study and could be removed.

We removed most of it but retained a suggestion that such SMS subduction could also influence the creation of subsurface phytoplankton maxima, as observed earlier in the Gulf of Finland during summer.

Missing Conclusion

• A summary of key findings and a clear answer to the hypotheses are needed.
• Clarify how glider data contributed—was it only for validation?
• Highlight the study's novelty and limitations clearly.

We formulated concluding remarks as follows: This study demonstrates that submesoscale variability in the Gulf of Finland is strongly modulated by both atmospheric forcing – particularly surface heat flux and wind stress – and background hydrographic structures such as mesoscale frontal gradients. Glider observations, supported by high-resolution modelling, revealed consistent spatial patterns of SMS activity, with spice anomalies

concentrated near the UML base in spring and within the thermocline in late summer, demonstrating the vertical sensitivity of SMS features to seasonal stratification. While seasonal stratification played a key role in shaping SMS structure, wind forcing became dominant under weaker surface buoyancy input. High spice variability and subduction signatures were consistently found on the offshore side of a baroclinic coastal current, where sloped isopycnals aligned with velocity and spice gradients indicated downward and lateral transport of surface-layer water masses. The integration of observations and model output allowed for extrapolation beyond individual glider transects, confirming that SMS processes in this coastal sea are both dynamically active and responsive to variations in external forcing. Together, these results clarify the physical mechanisms driving SMS variability and subduction in stratified coastal environments.

Recommendation: Major Revisions

The manuscript addresses a relevant topic and includes valuable datasets. However, substantial improvements in structure, clarity, and analysis are needed. The roles of observations and model outputs must be better defined, and the analysis should be aligned more closely with the stated objectives.

**References**

Klingbeil, K., Lemarié, F., Debreu, L., and Burchard, H.: The numerics of hydrostatic structured-grid coastal ocean models: State of the art and future perspectives, Ocean Modelling, 125, 80–105, doi:10.1016/j.ocemod.2018.01.007, 2018.

Wong, A., Keeley, R., Carval, T., and the Argo Data Management Team (2025). Argo Quality Control Manual for CTD and Trajectory Data. http://dx.doi.org/10.13155/33951

---

## Author Response (AR1)

**Author's Response to Reviewers**

Manuscript Title: Forcing-dependent submesoscale variability and subduction in the coastal sea area (Gulf of Finland, Baltic Sea)

Authors: Kai Salm, Germo Väli, Taavi Liblik, and Urmas Lips

We thank both reviewers for their thoughtful and constructive feedback. In response, we have made extensive revisions to improve the clarity, structure, and scientific rigor of the manuscript. Key changes include rewriting large sections for clarity and flow, improving the presentation of the study's motivation and novelty, clarifying definitions of key concepts, strengthening the analysis with new quantitative metrics and figures, and adding a dedicated Conclusion section. We believe these changes address all major concerns and significantly improve the manuscript's clarity and scientific value.

A lot of the reviewers' comments converge on similar aspects of the manuscript that have now been addressed.

**General Comments**

We acknowledged that language, grammar, and structure needed thorough revisions throughout the manuscript.

We have substantially revised the Introduction to clarify the narrative and strengthen the study's motivation and context. The updated text provides a clearer explanation of the relevance of submesoscale processes and the use of spice to characterise them in stratified estuarine systems, such as the Gulf of Finland. We also highlight one aspect of the novelty of our study in terms of combining glider-based observations with a regional high-resolution model to investigate the vertical structure and evolution of spice anomalies, which has not been previously addressed in the Gulf of Finland context or similar basins.

We have revised the manuscript to introduce key concepts, such as spice, tracer variance, Rossby number, and buoyancy gradients, earlier in the text, with clearer definitions and more consistent terminology throughout.

Use of Observations vs. Model: We have published an earlier paper based solely on glider data from 2018. We incorporated two more glider missions from the study area to characterize SMS variability in relation to forcing and mesoscale background and used model data to extrapolate (generalize) the findings over a larger spatial area and temporal extent. While the numerical model provides the spatial and temporal coverage necessary to explain the development of submesoscale processes, we have revised the manuscript to better emphasize the role of glider observations beyond validation. Specifically, we state that glider observations revealed maxima of spice above the maximum vertical density gradient during spring missions and below it during the late summer mission. Also, glider data indicated high spice in the sub-surface layer in the case of forcing conditions favourable for coastal upwelling and formation of a long-coastal baroclinic current. Both of these findings were generalized, and potential mechanisms were suggested using model data from a larger area and extended periods.

Figures: We have revised several figures to improve clarity and comparability, for example, by harmonizing axes across related plots and adding subplots showing vertical velocity and horizontal currents. In response to reviewer comments, new figures have been introduced, which has changed the figure sequence compared to the previous version. These enhancements provide a clearer view of subduction processes and allow for a more effective assessment of the physical mechanisms discussed in the manuscript.

The Discussion section has been reorganized to directly reference relevant figures and results and includes a more focused comparison with previous studies. These changes improve the cohesion of the manuscript and highlight how our findings contribute to understanding submesoscale dynamics and vertical structure in the GoF.

We have added a Conclusion section that clearly summarizes the main findings and implications of the study.

**Specific Comments:**

**Abstract**

We have reformulated the abstract to stress better the main findings and differentiate what is well founded by data and what can be suggested but must be studied in more detail in the future as was suggested by the reviewers.

**Introduction**

Both reviewers noted that the Introduction lacked clarity, structure, and clear motivation. In response, we thoroughly revised the section to improve grammar, sentence flow, and paragraph organization. We now emphasize the importance of submesoscale processes from the outset and clarify their links to key physical and biogeochemical processes. Key concepts such as SMS dynamics, spice, and tracer variability are now introduced earlier, defined more clearly, and connected to the study objectives. We restructured the Introduction to present a logical narrative: starting from the relevance of SMS flows, moving through the regional context of the Gulf of Finland, and leading to the rationale for combining glider observations and high-resolution modeling. We also clarified why gliders are used, the mission timeframes, and the advantages and limitations of the modeling approach. Hypotheses are now clearly formulated to guide the study and are tested in the Results. Collectively, these revisions provide a clearer background and stronger justification for the work.

**Methods**

Both reviewers requested clarification and improvement in the description of glider missions and data processing. In response, we now explicitly state the exact dates, transect orientation, and depth coverage of each glider mission. While originally conducted for different objectives, the missions collectively form the observational basis of this study. We clarified why transect orientations varied and included new details on the vertical profiling behavior of the glider.

Reviewer 1 asked for a description of data quality control procedures and interpolation methods; we now specify that quality control followed adapted Argo protocols and that interpolation was performed in time-pressure coordinates with a vertical resolution of 0.5 dbar and a horizontal resolution of 10 minutes. We also explained the rationale for time-based interpolation (raised by Reviewer 2), noting that it aligns with the glider's time-indexed sampling method.

Terminology was revised for clarity, including replacing "YOs" with "up- and downcasts" (Reviewer 2), and we clarified the model's adaptive vertical coordinate system and vertical interpolation of boundary conditions (Reviewer 1). Tense inconsistencies were corrected throughout. In response to a comment on Figure 1 (Reviewer 1), we slightly extended the map for better regional context.

**Analysis**

Both reviewers asked for justification of the 4 km horizontal averaging scale used for glider-derived spice. In response, we now clarify that this scale aligns with the internal Rossby deformation radius in the Gulf of Finland (Alenius et al., 2003) and offers a balance between preserving submesoscale signals and suppressing noise. To assess the sensitivity of our results to this choice, we provided additional figures 1 and 2 comparing spice distributions using 2 km, 4 km, and 7 km averaging. These demonstrate that key frontal features are robust across scales, though amplitude varies with smoothing.

Figure 1 shows an example of spice distribution using horizontal averaging scales of 2 km (a), 4 km (b), and 7 km (c). Each panel displays the same section, overlaid with density contours at 0.2 kg m-3 intervals. The data are based on glider observations from 24–25 May 2018.

Figure 2 shows the standard deviation of spice calculated from the glider mission conducted between 9 May and 6 June 2018. The colours indicate the horizontal averaging scales used for spice: 2 km (blue), 4 km (black), and 7 km (red).

Reviewer 2 requested that the spice equation be set apart for clarity, and a citation for TEOS-10 was added. Reviewer 1 asked us to specify how vertical buoyancy gradients  $(N^2)$  were computed; we now clarify that they were calculated using 2 m intervals. In response to both reviewers' suggestions, we harmonized axes in related figures to improve comparability, including converting spice plots to depth space where applicable.

Additional reviewer 1 requests included explaining the use of a 6-hour Gaussian filter on wind components (now clarified as a means to reduce high-frequency noise), and rephrasing or moving sentences that previously introduced results too early (now relocated to the Results section). Reviewer 2 suggested using glider data to estimate more SMS characteristics; we note that while possible, model output offers more continuous spatial and temporal coverage. This complements our earlier glider-only work (Salm et al., 2023).

**Results**

Both reviewers emphasized the need for clearer presentation and more quantitative comparisons between observations and model output. In response, we revised the language throughout the Results section for clarity and corrected missing or ambiguous terms (e.g., replacing "slight" with numerical differences). We now provide standard deviations alongside mean values (e.g., for UML, CIL, and maximum N² depths), and specify depths at which the model fails to capture observed secondary maxima. Reviewer 2's concern about the significance of model—observation mismatches has been addressed by clearly stating the magnitude of differences and their potential implications.

The figures were revised to better match the glider and model domains, remove gaps, and harmonize axes (e.g., depth vs. density space). We replaced mission-averaged spice fields with more representative cross-sections (Figs. 3 and 4 in revised version), and added markers to figures to highlight key events and periods referenced in the text.

Both reviewers asked for improved treatment of submesoscale indicators and a more quantitative discussion. Accordingly, subsection 3.3 was revised extensively to add quantifications where possible, and a new subsection 3.4 was developed to expand on specific events. Although some conclusions remain qualitative (e.g., the influence of coastal currents and topography on offshore spice maxima), we now support them with additional figures and mechanistic discussion. While a full quantitative attribution remains difficult, we clearly state the limitations and avoid overinterpreting model outputs.

**Discussion**

Both reviewers found that some parts of the Discussion were too general or disconnected from the study's findings. In response, we revised the section to focus more tightly on results from the present study and removed or rephrased general or speculative statements (e.g., Lines 367–374, 400–403, and 444–452 in previous version). Reviewer 1's concern about referencing vertical velocities was addressed by including vertical velocity plots in the revised figures (Figs. 11 and 13). Reviewer 2 requested clarification on "elongated regions" of spice; we specified that these are horizontal structures and added figure references.

Both reviewers questioned the inclusion of winter SMS dynamics, noting that the study only covers spring—summer. We agreed and removed related content to avoid confusion about the study's scope. We also now explicitly state a key methodological limitation: spice, as used here, is most appropriate during seasons when both temperature and salinity significantly contribute to density. Reviewer 1's question about strong claims based on limited cases led us to soften language and clarify that our conclusions are drawn from specific events rather than broader statistics. Where relevant, we now indicate whether statements are literature-based or derived from our results.

Overall, the Discussion has been streamlined, more directly tied to the presented findings, and refined to better reflect the study's seasonal scope and methodological constraints.

**Conclusion**

Both reviewers noted the absence of a dedicated conclusion summarizing the key findings. In response, we added a Conclusion section that synthesizes the main results, responds to the reviewers' expectations and strengthens the manuscript's overall structure.

---

## Referee Report (RR1)

I thoroughly enjoyed reading this revised manuscript, and would like to both thank and congratulate the authors for the excellent and rigorous work on the original paper. The paper is appropriate for publication "as-is", but as very very minor comments/corrections please see the list below – all only relevant for the introduction.

Line 25: "upper ocean's" to "upper ocean"

Line 27: "is" to "are"

Line 30: "SMS dynamics is" to "SMS dynamics are"

Lines 65-90: could be cut down on (I enjoyed reading them, but there's a lot of info there about what the paper is about to contain) – optional comment!